# Intercomparison of lidar, aircraft, and surface ozone measurements in the San Joaquin Valley during the California Baseline Ozone Transport Study (CABOTS)

Andrew O. Langford[1], Raul J. Alvarez II[1], Guillaume Kirgis[1,2], Christoph J. Senff[1,2], Dani Caputi[3], Stephen A. Conley[4], Ian C. Faloona[3], Laura T. Iraci[5], Josette E. Marrero[5,*], Mimi E. McNamara[5,7,§] , Ju-Mee Ryoo[5,‡], and Emma L. Yates[5,6]

[1]NOAA Earth System Research Laboratory/Chemical Sciences Division, Boulder, CO 80305, USA.
[2]Cooperative Institute for Research in Environmental Sciences, University of Colorado, Boulder, CO, 80309, USA.
[3]Department of Land, Air, and Water Resources, University of California, Davis, CA, 95616, USA.

[4]Scientific Aviation, Inc., Boulder, Colorado, 80301, USA.

[5]Atmospheric Science Branch, NASA Ames Research Center, Moffett Field, CA, 94035, USA.

[6]Bay Area Environmental Research Institute, Petaluma, CA 94952, USA.
[7] Environmental Science and Policy Department, University of California, Davis, CA, 95616, USA.

* Now at: Sonoma Technology, Inc., Petaluma, CA, 94954
§ Now at: Illingworth & Rodkin, Inc., Petaluma, CA 94954
‡ Now at: Science and Technology Corporation, Moffett Field, CA, 94035

*Correspondence to: Andrew O. Langford ([andrew.o.langford@noaa.gov](mailto:andrew.o.langford@noaa.gov))*

**Abstract.** The California Baseline Ozone Transport Study (CABOTS) was conducted in the late spring and summer of 2016 to investigate the influence of long-range transport and stratospheric intrusions on surface ozone ($O_3$) concentrations in California with emphasis on the San Joaquin Valley (SJV), one of two "extreme" ozone non-attainment areas in the U.S. One of the major objectives of CABOTS was to characterize the vertical distribution of $O_3$ and aerosols above the SJV to aid in the identification of elevated transport layers and assess their surface impacts. To this end, the NOAA Earth System Research Laboratory (ESRL) deployed the Tunable Optical Profiler for Aerosol and oZone (TOPAZ) mobile lidar to the Visalia Municipal Airport (36.315°N, -119.392°E) in the central SJV between 27 May and 7 August 2016. Here we compare the TOPAZ ozone retrievals with co-located *in-situ* surface measurements and nearby regulatory monitors, and to airborne *in-situ* measurements from the University of California at Davis/Scientific Aviation (SciAv) Mooney and NASA Alpha Jet Atmospheric eXperiment (AJAX) research aircraft. Our analysis shows that the lidar and aircraft measurements agree, on average, to within 5 ppbv, the sum of their stated uncertainties of 3 and 2 ppbv, respectively.

**1 Introduction**

The San Joaquin Valley (SJV) of California is one of only two "extreme" ozone ($O_3$) non-attainment areas remaining in the United States with a 2016 ozone Design Value, i.e. the metric used by the U.S. EPA to determine air quality compliance that is calculated as the 3-yr average of the $4^{th}$ highest measured maximum daily 8-h average mixing ratio (MDA8), that is more than 20 parts-per-billion by volume (ppbv) greater than the primary National Ambient Air Quality Standard (NAAQS) of 70 ppbv (https://www3.epa.gov/airquality/greenbook/hdtc.html). Such high $O_3$ concentrations are harmful to human health (U.S. Environmental Protection Agency, 2014) and impair plant growth and productivity (Avnery et al., 2011a, b), adversely affecting both the $15 billion agricultural industry in the SJV and the iconic forests of the nearby Sequoia and Kings Canyon National Parks (Panek et al., 2013).

The need to better understand the causes for the high surface $O_3$ in the San Joaquin Valley has motivated several major air quality studies over the years including the San Joaquin Valley Air Quality Study (SJVAQS) in 1990 (Lagarias and Sylte, 1991), the Central California Ozone Study (CCOS) in 2000, (Reynolds et al., 2010) and the California Research at the Nexus of Air Quality and Climate Change (CalNex) field campaign in 2010 (Ryerson et al., 2013; Brune et al., 2016). More recently, this issue was addressed by the 2016 California Baseline Ozone Transport Study (CABOTS) organized and supported by the California Air Resources Board (CARB) (https://www.arb.ca.gov/research/cabots/cabots.htm). CABOTS was designed to investigate the contributions of background $O_3$ (Jaffe et al., 2018) and the influence of stratospheric intrusions (Lin et al., 2012a) and long-range transport from Asia (Lin et al., 2012b) on surface $O_3$ concentrations in the SJV during late spring and summer. Characterization of the vertical distribution of $O_3$ in the lower and middle free troposphere above the SJV and upwind regions with an accuracy of at least 10%, the nominal accuracy of ECC ozonesondes in the troposphere (Smit, et al., 2014), was a key objective of the campaign, and $O_3$ profiles were measured using three different techniques (lidar, aircraft, and ozonesondes) in various parts of California. Integration of these datasets requires that these measurements be intercompared (Ancellet and Ravetta, 2005; Beekmann et al., 1995; Kempfer et al., 1994; Schäfer et al., 2002) and any differences between the various techniques understood and characterized. For pollution studies, it is important that this validation includes the lowest 100 m, which is inaccessible to most ozone lidars (Wang et al. 2017). In this paper, we compare $O_3$ measurements from the NOAA ESRL multi-angle Tunable Optical Profiler for Aerosol and oZone (TOPAZ) lidar with *in-situ* measurements from nearby regulatory and research surface monitors, and from instruments flown aboard the UC Davis/Scientific Aviation Mooney (Trousdell et al., 2016) and Alpha Jet research aircraft based at NASA's Ames Research Center (Hamill et al., 2016;Yates et al., 2015). These comparisons, together with those from the multi-lidar (including TOPAZ) and ozonesonde Southern California Ozone Observation Project (SCOOP) intercomparison conducted by the NASA-sponsored Tropospheric Ozone Lidar Network (TOLNet) immediately after CABOTS (Leblanc et al., 2018), provide this validation.

**2 California Baseline Ozone Transport Study (CABOTS)**

The CABOTS field campaign was conducted between mid-May and mid-August of 2016. The primary measurements (cf. **Figure** 1a) included electrochemical cell (ECC) ozonesondes (Johnson et al., 2002) launched

daily from Bodega Bay (38.319°N, -123.075°E, 12 m asl) (6 May-17 August) and Half Moon Bay (37.505°N, -122.483°E, 9 m asl) (15 July-17 August) by the San Jose State University (SJSU), *in-situ* aircraft sampling of $O_3$ and other compounds above central California by the University of California, Davis (UC Davis)/Scientific Aviation (Trousdell et al., 2016) and the NASA Alpha Jet Atmospheric eXperiment (AJAX) (Yates et al., 2015), and ozone and backscatter lidar measurements by the truck-based NOAA ESRL TOPAZ lidar system (Alvarez et al., 2011) at the Visalia Municipal Airport (VMA, 36.315°N, -119.392°E, 88 m above mean sea level, asl) (27 May-18 June and 18 July-7 August) (**Figure** 2). Surface $O_3$ measurements were also made at the ozonesonde and lidar sites, and at the UC Davis monitoring station at the Chews Ridge Observatory (36.306°N, -121.567°E, 1520 m asl) (Asher et al., 2018) in the Santa Lucia Mountains west of Visalia, as well as the extensive networks of regulatory surface monitors maintained by the California Air Resources Board and the San Joaquin Valley Unified Air Pollution Control District (SJVAPCD).

The Bodega Bay and Half Moon Bay sites were located on the coast to sample the Pacific inflow, and the VMA was chosen for the TOPAZ operations because of its central location in the SJV, the availability of the runway and airspace for low approaches and aircraft profiles, and the presence of the co-located SJVAPCD wind profiler and Radio Acoustic Sounding System (RASS) (Bao et al., 2008). The TOPAZ truck was parked on the west side of the VMA between the airport runway and the heavily-trafficked multi-lane CA-99 and adjacent San Joaquin Valley Railroad (SJVR) (**Figure** 2). The VMA is located about 10 km west of downtown Visalia (pop. 130,000) and lies about one-third (60 km) of the way from Fresno to Bakersfield (**Figure** 1a,b). Visalia is located about 400 km from Bodega Bay, and 300 km from Half Moon Bay, which limited the usefulness of comparisons between the lidar and ozonesondes.

**3 Ozone Measurement Platforms**

**3.1 NOAA/ESRL TOPAZ lidar**
The TOPAZ differential absorption lidar (DIAL) system was originally developed for the profiling of $O_3$ and particulate backscatter in the planetary boundary layer and lower free troposphere from NOAA Twin Otter aircraft (Alvarez et al., 2011;Langford et al., 2011;Senff et al., 2010;Langford et al., 2012;Langford et al., 2010). The lidar was reconfigured for mobile ground-based measurements in 2012 and deployed in this configuration to several field campaigns including the 2013 Las Vegas Ozone Study (LVOS) (Langford et al., 2015) prior to CABOTS. The lidar is installed in the back of a medium box truck (cf. **Figure** 2) equipped with a commercial UV absorption monitor for *in-situ* $O_3$ measurements (2B Technologies Model 205) that samples air 5 m above the surface and an Airmar 150WX weather station to measure temperature, pressure, relative humidity, and wind speed and direction. The 2B Model 205 has been approved by the EPA as a Federal Equivalent Method (FEM) for surface $O_3$ monitoring and has a nominal ($1\sigma$) precision and accuracy that is the greater of 1 ppbv or 2% for 10-s averages. Modified versions of the same instrument were flown on both the Scientific Aviation Mooney and NASA Alpha Jet. Comparisons

between the NOAA 2B at the VMA and a mobile calibration source operated by CARB revealed a 3% low bias in the recorded 2B measurements that has been corrected in the data used here.

The eye safe TOPAZ lidar is built around a low pulse energy (~100 μJ), high repetition rate (1 kHz) quadrupled Nd:YLF pumped Ce:LiCAF laser that is re-tuned between each pulse to generate light at three different wavelengths from 286 to 294 nm with an effective repetition rate of 333 Hz for each wavelength (Alvarez et al., 2011). The laser pulses are transmitted and the lidar return signals collected by a coaxial transmitter/receiver equipped with a commercial (Licel) photomultiplier-based dual analog/photon counting system. This hybrid data acquisition system was installed in 2016 and replaced the original fast analog data acquisition system that was optimized for aircraft operations (Alvarez et al., 2011;Wang et al., 2017). This modification increased the maximum useful range to ~6 km during the day and to more than 8 km at night, depending on the laser power, atmospheric extinction, and solar background light.

The truck-mounted version of TOPAZ incorporates a large scannable turning mirror above the vertically pointing transmitter/receiver to allow profile measurements at different slant angles. These slant profiles can be combined to create vertical profiles that start much closer to the ground (25-30 m) than conventional vertically staring lidar systems (Proffitt and Langford, 1997). During CABOTS, the scanning mirror was moved sequentially between elevation angles of 90, 20, 6, and 2° with a 225-s averaging time at 90° and 75-s averaging times at the other 3 angles. The cycle was repeated approximately every 8 minutes and the vertical projections combined to create a single vertical profile starting at 27.5±5 m above ground level (agl). This approach assumes a fair degree of horizontal homogeneity and the lidar slant paths were oriented parallel to the VMA runway (135°) over open farmland to avoid populated neighborhoods and minimize the effects of NO emissions from the often heavy traffic on CA-99 (cf. **Figure** 2), which could locally titrate ozone and create strong horizontal concentration gradients near the surface.

The $O_3$ profiles shown here were retrieved using two wavelengths (~287 and 294 nm) with 30-m range gates and a smoothing filter that increased from 270 m wide at the minimum range (815±15 m) to 1400 m wide at the maximum range (8 km). The effective vertical resolution increased from ~10 m near the surface to ~150 m above 500 m agl and 900 m at 6 km. Profiles of the backscatter from aerosols, smoke, and dust were retrieved with a constant 7.5 m resolution at 294 nm. The ozone profiles were computed using the $O_3$ absorption cross-sections from Malicet *et al.* (1995) and an iterative technique to correct for differential aerosol backscatter and extinction that assumes a backscatter-to-extinction ratio of 40 and fixed Ångstrom coefficients of 0 for backscatter and -0.5 for extinction (Alvarez et al., 2011). These values offer a good compromise for a wide variety of particulate types (Völger et al., 1996). The actual aerosol composition in the SJV was not measured during CABOTS, but measurements during the 2010 Carbonaceous Aerosols and Radiative Effects Study (CARES) typically found a mix of organics, sulfate, nitrate, ammonium, and soil dust in the northern part of the valley (Zaveri et al., 2012). Smoke from the Soberanes Fire near Big Sur dominated the aerosol mix in the SJV during the second IOP. We varied the aerosol backscatter

Angstrom coefficient between -1 and 1 and the aerosol extinction Angstrom coefficient between 0 and -1 for a
"worst case scenario" of a thin smoke layer with very high aerosol backscatter embedded in an otherwise clean
atmosphere to estimate the error in the ozone retrieval introduced by using these fixed parameters. The sharp aerosol
gradients at the smoke layer edges tend to magnify errors in the ozone retrieval if the aerosol correction is not
properly implemented. Temperature and pressure profiles interpolated from the 3-h National Centers for
Environmental Prediction (NCEP) North American Regional Reanalysis (NARR) using the grid point closest to the
TOPAZ lidar location were used to account for the temperature dependence of the $O_3$ cross-sections and to convert
$O_3$ number densities to mixing ratios. The total uncertainties in the 8-min ozone retrievals in the absence of strong
aerosol gradients are estimated to increase from ±3 ppbv below 4 km to ±10 ppbv at the top of the profile. When
strong backscatter gradients are present, the $O_3$ uncertainty can potentially increase by another ±3 ppbv.
**3.2 UC Davis/Scientific Aviation Mooney**
The University of California at Davis and Scientific Aviation, Inc. (http://www.scientificaviation.com), conducted a
series of research flights above the SJV during the summer of 2016 using a Scientific Aviation single-engine
Mooney TLS or Ovation aircraft as part of the CARB-supported Residual Layer Ozone Study (RLO)
(https://www.arb.ca.gov/research/apr/past/14-308.pdf). Several of these flights overlapped with the TOPAZ
operations during CABOTS, as did some of the 12 additional flights (EPA) funded by the U.S. EPA and the Bay
Area Air Quality Management District (BAAQMD). The Mooney carried a 2B Technologies Model 205 $O_3$
monitor, an Eco Physics Model CLD 88 (NO) with a photolytic converter to measure NO and $NO_2$, and a Picarro
2301f Cavity Ring-Down Spectrometer (CRDS) to measure $CO_2$, $CH_4$, and $H_2O$ (Trousdell et al., 2016). The 2B
model 205 was used with the minimum integration time of 2 s, which corresponds to a mean distance of 150 m at
the typical level flight speed (the data stream was sampled at 1-s intervals). As noted above, the 2B has a nominal
accuracy of 2% for concentrations above 5 ppbv, and a precision of 2% for concentrations above 5 ppbv if 10-s
averages are used. If the limiting noise is randomly distributed, this implies a precision of 5 ppbv for 2-s averages.
Calibrations of the Scientific Aviation 2B using an external ozone source (2B, Model 306) found the instrument to
have offsets and slopes less than 1.5 ppb and within 4% of unity, respectively.
**3.3 NASA Alpha Jet Atmospheric eXperiment (AJAX)**
The NASA Ames Alpha Jet Atmospheric eXperiment (AJAX) (Hamill et al., 2016) sampled $O_3$ and other
tropospheric constituents above California during CABOTS using a two-person jet based at Moffett Field, CA (MF,
37.415° N, -122.050° E). The Alpha Jet carried an external wing pod with a modified commercial UV absorption
monitor (2B Technologies Inc., model 205) to measure $O_3$ (Ryoo et al., 2017;Yates et al., 2015;Yates et al., 2013)
and a (Picarro model 2301-m) cavity ringdown analyzer to measure $CO_2$, $CH_4$, and $H_2O$ (Tanaka et al., 2016). A
second wing pod carried a non-resonant laser-induced fluorescence instrument to measure formaldehyde ($CH_2O$)
(St. Clair et al., 2017). The pod mounting kept the residence times of the sample inlets to less than 2 s. The aircraft is
also equipped with GPS and inertial navigation systems to provide altitude and position information, and the NASA
Ames-developed Meteorological Measurement Systems (MMS) to provide highly accurate pressure, temperature,
and 3-D wind data. The 2B $O_3$ data, recorded every 2 s, are averaged over 10 s to increase the signal-to-noise ratio.
Ozone calibrations were performed before/after each flight using an external ozone source (2B Technologies Inc.,
model 306 referenced to the NIST scale, certified annually). Raw flight $O_3$ data were corrected using the linearity
correction factor and zero offset from the calibration closest in time to the flight. Overall accuracy of the $O_3$
instrument is determined to be 3 ppbv or better at 10-s resolution, with uncertainty improving at lower altitudes, as
determined from pressure chamber tests; see Yates et al., (2013) for a more detailed error analysis.
**4 Results and Comparisons**
The TOPAZ measurements were conducted over two 3-week intensive operating periods (IOPs) in the late spring
(27 May to 18 June) and summer (18 July to 7 August) of 2016. A total of 440 hours of lidar data were recorded
during the first (1654 profiles over 22 days) and second (1686 profiles over 21 days) IOPs with an average of more
than 10 hours of nearly continuous measurements per day. The skies above Visalia were mostly cloud free during
the study, with only a few profiles truncated by high clouds during IOP1. However, during IOP2 the SJV was
fumigated by smoke from the Soberanes Fire that started on 22 July about 200 km west of Visalia near Big Sur.
**4.1 Comparisons between lidar and surface measurements**
The NOAA 2B ozone monitor operated continuously at the VMA throughout the TOPAZ deployment with the
system response checked during each IOP by an external mobile calibration source operated by CARB. These
calibration checks revealed a 3% low bias in the NOAA 2B instrument that has been corrected in the data shown
here. **Figure 3** plots time series (Pacific Daylight Time, PDT, or UT-7 h) of the 1-min averaged *in-situ* surface
mixing ratios (gray dots) measured 5 m above the ground from each IOP together with the TOPAZ mixing ratios
retrieved from a height of 27.5±5 m (black line) and a range of 815±15 m along the slant path above the agricultural
fields to the southeast (cf. **Figure** 2). **Figure** 4a is an enlarged view of the VMA surface measurements (gray line)
from 9-13 June together with the mixing ratios from the 27.5 m TOPAZ measurements (filled black circles). Also
plotted are the 1-h average ozone mixing ratios measured 6.7 m agl by the CARB regulatory API/Teledyne 400
monitor located on N. Church Street in Visalia (102 m asl) about 10 km to the east of VMA (solid black line), and
measured 5 m agl by the SJVAPCD API/Teledyne 400 monitor in Hanford (82 m asl) about 22 km to the west of
VMA (dotted black line). The four sets of measurements agreed fairly well during the day but diverged markedly at
night and in the early morning when $O_3$ was removed by surface deposition and titration by $NO_x$ within the surface
layer. The losses were greatest at the VMA monitor which was located in the TOPAZ truck next to the heavily-
trafficked CA-99 and SJVR railroad line. Titration by NO was undoubtedly much greater here, but there were no
NOx measurements available to confirm this hypothesis. Much smaller losses were measured by the rural Hanford
monitor and intermediate losses were measured by the Visalia monitor which is located on a downtown rooftop. A
scatter plot of all of the coincident TOPAZ and in-situ measurements from CABOTS (**Figure 4**b, filled gray circles)
shows that the in-situ concentrations measured at VMA were often much smaller than the concentrations measured
815±15 m away by the lidar, and even titrated to zero under some conditions. The data converge (filled black

circles) when the comparison is restricted to conditions when the two measurements are expected to sample a common airmass, i.e. during the day after the nocturnal inversion has dissipated (0900 to 1830 PDT) and the winds were southeasterly (125 to 145°) and greater than 2.5 m s$^{-1}$. The results of Orthogonal Distance Regression (ODR) fits of these data are shown both in the figure and in **Table** 1. We use ODR fits, which assume that both variables can have uncertainties, for our analyses instead of simple linear regressions which assume that all of the uncertainties lie in the dependent variable. Fits of the filtered data give a slope of 1.00±0.03 and an intercept of - 2.6±1.5 ppbv where the errors represent the 95% confidence limits of the ODR fits.

**Figure** 5 compares the 27.5 m TOPAZ O$_3$ measurements to the regulatory O$_3$ surface measurements from the monitors at Visalia (8.5 km) and Hanford (24 km) described above, and from the more distant SJVAPCD monitors at Parlier (34 km) and Porterville (43 km). The TOPAZ mixing ratios were slightly higher than those at Visalia and Hanford, but lower than those at Parlier and Porterville, which are closer to the Sierra foothills and measure some of the highest O$_3$ concentrations found in the SJVAB. The degree of correlation decreased with distance as expected, yet remained quite good more than 40 km from the VMA at Porterville. This suggests that the O$_3$ measurements acquired at the VMA during CABOTS can be considered representative of the central San Joaquin Valley.

**4.2 Comparisons between lidar and aircraft measurements**

Comparisons between the ground-based lidar and aircraft measurements are subject to much larger uncertainties arising from spatial and temporal sampling differences compared to the comparison with nearby surface monitors. During CABOTS, the fixed wing aircraft conducted both low approaches above the VMA runway (cf. **Figure** 2) and spiral profiles around the airport, but never directly sampled the vertical column probed by the lidar. The comparisons were also conducted as brief elements of multi-hour sampling flights with other objectives, and time constraints and air traffic considerations sometimes contributed to the spatial and temporal mismatches. The piston-engine Mooney took about 25 minutes to execute an ascending profile from the surface to 3 km, while the Alpha Jet took about 9 minutes (similar to the 8-min TOPAZ integration time) to conduct a descending profile from 3 km to the surface. Spatial mismatches were also created by the vertically smoothing of the DIAL retrieval, which can both smooth and displace sharp vertical concentration gradients seen by the aircraft. Similar considerations apply to comparisons between lidars and ozonesondes since balloons have a finite rise time and can be carried many kilometers downwind from the launch site (Leblanc et al. 2018). Despite these caveats, we show that the lidar and aircraft measurements usually agreed to within ±10%, the nominal accuracy of ECC ozonesondes in the troposphere (Smit, et al., 2014), which is the generally accepted reference standard for ozone profile measurements.

**4.2.1 UC Davis/Scientific Aviation Mooney**

The RLO flights were executed as a series of 2 to 3-day deployments with as many as 4 flights per day lasting 2 to 3 hours each between Fresno and Bakersfield. Two of these deployments, RLO2 (2-4 June), and RLO4 (24-26 July), overlapped with the first and second TOPAZ IOPs, respectively, and included low approaches at VMA on most of the flights with spiral profiles near VMA on several. Both deployments occurred as warm temperatures (>40°C) and

weak anticyclonic winds associated with synoptic high-pressure systems resulted in the buildup of surface ozone across the South Coast and San Joaquin Valley Air Basins. The highest measured MDA8 $O_3$ in the SJVAB during the first IOP was recorded on 4 June at Clovis (91 ppbv), which lies about 65 km northwest of VMA (cf. **Figure** 1b). The highest reported MDA8 $O_3$ during the second IOP (and the year) was recorded on 27 July at Parlier (101 ppbv), which lies midway between Clovis and the VMA. The monitors at Visalia and Hanford reported MDA8 concentrations of 72 and 88 ppbv, respectively, on 4 June, and 83 and 85 ppbv on 27 July. **Figure** 3 shows that the highest $O_3$ mixing ratios measured by the VMA surface monitor and TOPAZ (27.5 m agl) were also recorded on these two days.

The flight tracks from all of the Mooney sorties during the RLO2 and RLO4 deployments are plotted in **Figure** 6a. FLT29 (RLO4) was a transit flight from the Scientific Aviation home base near Sacramento to Fresno. The remaining RLO flights were between Fresno and Bakersfield as noted above. The two EPA deployments (27-29 July and 4-6 August) were of longer duration than the RLO flights with morning and afternoon sorties that placed more emphasis on cross-valley measurements and transects to the coast (**Figure** 6b) including profiles above the South Bay (EPA1) and Chews Ridge (EPA2). The afternoon flights during both series included legs to Visalia.

**Figure** 7 shows the sections of the RLO and EPA flight tracks that passed within 5 km of TOPAZ (dashed black circles). Most of these flights included low (<10 m) passes along the VMA runway that approached to within ~350 m horizontally of the TOPAZ truck and within 1000 m of the center of the 27.5 m agl TOPAZ slant path measurements (cf. **Figure** 2). **Figures 8a-8d** show time series of the 27.5 m TOPAZ and 5 m *in-situ* measurements during all of the RLO and EPA low approaches together with the ozone measured by the aircraft between the surface and 25 m agl. All of the aircraft measurements lie within 10% of the $O_3$ retrieved by TOPAZ with the exception of the much higher values (>100 ppbv) measured by the Mooney around 1400 PDT on 3 June (**Figure 8**a, see below). The scatter plots in **Figures** 8e and 8f show that the aircraft also measured much higher concentrations than the *in-situ* surface monitor during the night and early morning, in agreement with the lidar measurements in **Figure** 4. The differences were smaller on 27 July than on 3 June, and also less pronounced than those in **Figure** 4. Closer agreement between the aircraft and surface measurements might be expected since some of the aircraft measurements were made within 200 m of the lidar truck (cf. **Figure** 2). The dark blue points show that the low bias in the surface measurements decreased during the day after the surface inversion had dissipated (there were too few measurements to effectively filter them by windspeed or direction). The mean ODR fit parameters based on the measurements from both RLO2 and RLO4 listed in **Table** 1 are very similar to those found for the lidar which suggests that the filtered surface measurements still have low bias that could be either instrumental or sampling related.

**Figure 9** compares the aircraft and lidar $O_3$ measurements made during 5 of the ascending profiles conducted by the Mooney near the VMA. FLT19 was conducted in the early afternoon of 3 June, and FLT33, FLT35, FLT36, and FLT37 were conducted over the 24-hour period beginning just after local midnight on 25 July. The four consecutive

TOPAZ profiles acquired during the time required for the Mooney to reach the top of each profile (~15-30 minutes at a climb rate of ~2.2 m s$^{-1}$) are plotted in each panel. The gray envelopes show the lidar mean profile ±10%. The differences between consecutive profiles reflect the combined effects of atmospheric variability and the precision of the lidar measurements.

Overall, the agreement between the TOPAZ and Mooney profiles in **Figure 9** is within ±10%, but there are some notable discrepancies. Most of these arise from the coarser vertical resolution of the lidar retrievals, which smooth out abrupt concentration changes such as those seen at the top of the boundary layer (~0.8 km agl) in **Figure 9**a, and between 2 and 3 km in **Figure 9**e where several narrower layers are smoothed into one broad layer in the lidar profile. **Figure 9**e also shows that the agreement between the lidar and aircraft measurements is better at low altitudes where the addition of the slant path measurements significantly improves the effective vertical resolution of the lidar. Fine-scale variability in $O_3$ also contributes to some of the observed differences, particularly on 3 June where the aircraft-measured $O_3$ concentrations varied by as much 25 ppbv during the low approach over the VMA runway. This unusually large variability is also seen in the large and rapid changes in the lidar measurements near the top of the boundary layer (**Figure 9**a) and challenges the assumptions about horizontal homogeneity used in the calculation of the TOPAZ vertical profiles near the surface.

The lidar profiles from 26 July (**Figure 9**e) also show large profile-to-profile changes in the narrow high $O_3$ layer lying just above the top of the nocturnal boundary layer (~0.3 km asl). The 25 and 26 July measurements (**Figures 9**b-9e) were made several days after the Soberanes Fire started and the low altitude "layer" near 400 m in **Figure 9**e is actually a short-lived puff of smoke and elevated $O_3$ from the fire. This is more obvious in the expanded view of the profiles shown in **Figure 10**a. Only two of the four lidar profiles from **Figure 9**e are plotted: the first profile coinciding with the aircraft measurements (solid trace, ±10%) and the profile acquired 16-24 minutes later when the puff had mostly disappeared (dashed trace). The corresponding lidar backscatter measurements are plotted in **Figure 10**b, and **Figure 10**c shows the $NO_2$ and $H_2O$ profiles measured by the aircraft. The backscatter measurements show that the TOPAZ retrievals are unaffected by strong backscatter gradients, which can create second-derivative like inflection points in the DIAL $O_3$ profiles (Kovalev and McElroy, 1994). The absence of a corresponding structure in the aircraft $NO_2$ and $H_2O$ profiles confirms that the high $O_3$ layer seen in the lidar and aircraft measurements was not an artifact caused by interferences from these species, which weakly absorb between 280 and 300 nm (Proffitt and Langford, 1997).

**4.2.2 NASA Alpha Jet Atmospheric eXperiment (AJAX)**

AJAX conducted 4 research flights over the SJV while TOPAZ was operational, with 2 additional flights (21 June and 7 July) between the two IOPs. The Alpha Jet executed descending spiral profiles from 4 to 5 km down to the surface that ended in low approaches on three of these flights: AJX190 on 3 June, AJX191 on 15 June, and AJX195 on 21 July. The aircraft also conducted a very low approach (~5 m) at VMA on 28 July (AJX196) but did not execute a full profile. These low approach measurements are represented by the filled yellow circles in Figures 8a

and 8c. The first and last flights (AJX190 and AJX196) coincided with the high ozone episodes mentioned earlier and the third flight (AJX195) also occurred during a period of high pressure. The second flight (AJX191) was conducted as a deep closed low moved into the Pacific Northwest, however, bringing unseasonably cool temperatures (26 °C) and strong surface winds to the SJV. This cyclonic system advected a large Asian pollution plume across the valley in the middle troposphere, but surface ozone remained low with the peak MDA8 $O_3$ concentration in the SJVAB only reaching 59 ppbv at the Sequoia-Kings Canyon monitor.

**Figures 11** and **12** are similar to **Figures** 6 and 7, but instead show the AJAX flight tracks. The first AJAX flight (AJX190) on 3 June during IOP1 overlapped with the UC Davis/Scientific Aviation RLO2 deployment. AJX191 took place about two weeks later in IOP1, and AJX195 occurred several days prior to the RLO4 deployment in IOP2. AJAX also executed profiles (not shown here) above and upwind of Chews Ridge on AJX190 and AJX191 and near Bodega Bay on AJX191 and 195 and sampled the Soberanes Fire plume on AJX196.

**Figure 13** displays coincident AJAX and TOPAZ profiles in plots similar to those shown for the Mooney in **Figure** 9, but with an extended vertical axis to reflect the higher range of these profiles. The points in **Figure 13** are sparser than those in **Figure 9** in part because of the 10-s averaging time, and in part because the Alpha Jet executed its descending profiles with an airspeed of about 110 m s$^{-1}$ compared to about 60 m s$^{-1}$ for the ascending Mooney profiles.

The agreement between the Alpha Jet and TOPAZ measurements is within ±10% on all 3 days except for 3 June (**Figure 13a**) when the measured aircraft and retrieved lidar concentrations differ by as much as 12 ppbv (20%) at 2.5 km asl and 20 ppbv (~50%) at 5.2 km asl. The disparities between the inbound and outbound measurements in **Figure** 13a show that the Alpha Jet encountered strong horizontal gradients below 800 m in the boundary layer when it arrived at the VMA about 3 hours after the Mooney found similar horizontal variability (cf. **Figures 8a** and **9**a). The Google Earth map and latitude-altitude and longitude-altitude plots in **Figure 14** better illustrate the extent of the horizontal variability in the boundary layer. These figures also show weaker horizontal gradients above 3 km where the disagreement between the lidar and aircraft is most pronounced.

**5 Discussion**

The results of the different $O_3$ comparisons are summarized in **Table** 1. As was noted above, comparisons between the lidar and aircraft profiles are subject to uncertainties arising from sampling differences introduced by the intrinsic vertical smoothing of the lidar retrievals and horizontal displacements between the aircraft and lidar. The potential impact of horizontal displacements on the comparisons when the $O_3$ spatial variability is large is illustrated by **Figure** 14, and a good example of the differences created by the lidar smoothing is seen near the top of the boundary layer around 0.8 km in **Figure** 9a. These uncertainties can be reduced by averaging the measurements to be compared over larger volumes. **Figure** 15 compares the lidar and aircraft measurements from the profiles plotted in **Figures** 9 and 13, and from several other RLO and EPA flights not shown, with each individual profile averaged

over 1 km segments (0 to 1 km, 1 to 2 km, etc.). This averaging decreases the influence of $O_3$ spatial variability, and also reduces the statistical uncertainties in both the lidar retrievals and aircraft measurements, with the effective temporal averaging of the AJAX and SciAv measurements increasing to about 2 and 4 minutes, respectively. Each point in the scatter plots of **Figure** 15a and 15b represents the mean mixing ratio from one of these 1 km segments, with the error bars showing the standard deviation of the mean. The intercepts and slopes derived from orthogonal distance regressions of both datasets overlap with zero and unity, respectively, within the 95% confidence limits of the ODR fits. The lower panels (**Figures** 15c and 15d) plot the same data as differences which show that the TOPAZ and SciAv measurements (**Figure** 15c) agree to within 1 ppbv on average, and the TOPAZ and AJAX measurements (**Figure** 15d) to within 4.2 ppbv. Neither plot shows evidence of a systematic altitude dependence in the differences.

Both lidar/aircraft comparisons are limited by the small number of common measurements with only 3 profiles available for the AJAX comparisons. The SciAv comparisons include data from 7 flights, but only the 5 profiles shown in **Figure** 9 extend above 2 km and only 3 of those reach 3 km. These limited datasets make the comparisons more sensitive to the influence of individual points. For example, the point surrounded by the dashed circle in **Figure** 15d includes the measurements from within the dashed oval in **Figure** 13b where the lidar retrieval is clearly smoothing out vertical gradient compared to the aircraft measurements. If this measurement point is excluded, the mean TOPAZ-AJAX difference decreases to 3.9±2.6. In either case, the differences between the TOPAZ lidar retrievals and the in-situ surface and aircraft measurements lie within the combined uncertainties of the different measurements and well within the 10% accuracy standard set by the ECC ozonesonde.

**6 Summary and Conclusions**

The lidar, aircraft, and ozonesonde profiles acquired during the 2016 CABOTS field campaign provide an unprecedented look at the vertical distribution of lower tropospheric $O_3$ above California during late spring and summer. The good agreement between the low elevation TOPAZ measurements and the collocated and regional (<45 km) surface monitors suggests that the measurements made at the VMA during CABOTS can be considered representative of the central San Joaquin Valley. Comparisons between the NOAA TOPAZ lidar profiles and the surface and aircraft measurements agree within the stated uncertainties, and we conclude that all of these $O_3$ measurements may be used with confidence.

The coordinated lidar and aircraft sampling of $O_3$ above the central San Joaquin Valley during CABOTS also illustrates the synergy between the two types of measurements. Lidar can provide long time series of the $O_3$ (and backscatter) vertical distributions above a fixed location while the aircraft can place the lidar measurements within a larger spatial context and measure other important parameters. This synergy is illustrated by the two time-height curtain plots displayed in **Figure** 16. **Figure** 16a shows the continuous TOPAZ measurements from a 14-hour time span on 25-26 July with the data from SciAv FLT 35, 36, and 37 superimposed. The aircraft measurements made

within 5 km of VMA are highlighted by colored squares outlined in white. **Figure 16b** is similar, but shows 10-
hours of continuous TOPAZ measurements from 15 June with the AJAX measurements (AJX191) superimposed.
The CABOTS ozonesondes were launched too far away (>300 km) from the VMA to allow quantitative
comparisons with the lidar. However, TOPAZ was relocated to the NASA Jet Propulsion Laboratory (JPL) Table
Mountain Facility (TMF) in the San Gabriel Mountains immediately after CABOTS for the Southern California
Ozone Observation Project (SCOOP), a multiple lidar and ozonesonde intercomparison organized by the NASA-
sponsored Tropospheric Ozone Lidar Network or TOLNet (https://www-air.larc.nasa.gov/missions/TOLNet/) at the
NASA Jet Propulsion Laboratory (JPL) Table Mountain Facility (TMF) (Leblanc et al., 2018). The results from the
SCOOP intercomparison and those presented here complete the inter-validation of the CABOTS lidar, aircraft, and
ozonesonde profile measurements.
**Acknowledgements**
The California Baseline Ozone Transport Study (CABOTS) field measurements described here were funded by the
California Air Resources Board (CARB) under contracts #15RD012 (NOAA ESRL), #14-308 (UC Davis), and
#17RD004 (NASA Ames). We would like to thank Jin Xu and Eileen McCauley of CARB for their support and
assistance in the planning and execution of the project, and are grateful to the CARB and the San Joaquin Valley
Unified Air Pollution Control District (SJVAPCD) personnel who provided logistical support during the execution
of the field campaign. We would also like to thank Cathy Burgdorf-Rasco of NOAA ESRL and CIRES for
maintaining the CABOTS data site. The NOAA team would also like to thank Ann Weickmann, Scott Sandberg,
and Richard Marchbanks for their assistance during the field campaign. The NOAA/ESRL lidar operations were
also supported by the NOAA Climate Program Office, Atmospheric Chemistry, Carbon Cycle, and Climate (AC4)
Program and the NASA-sponsored Tropospheric Ozone Lidar Network (TOLNet, http://www-
air.larc.nasa.gov/missions/TOLNet/). The UC Davis/Scientific Aviation measurements were also supported by the
U.S. Environmental Protection Agency and Bay Area Air Quality Management District through contract #2016-129.
I.C.F. was also supported by the California Agricultural Experiment Station, Hatch project CA-D-LAW-2229-H.
The NASA AJAX project was also supported with Ames Research Center Director's funds, and the support and
partnership of H211, LLC is gratefully acknowledged. J.E.M. and J.-M.R. were supported through the NASA
Postdoctoral Program, and M.E.M. was funded through the Center for Applied Atmospheric Research and
Education (NASA MUREP). The CABOTS data are archived at https://www.esrl.noaa.gov/csd/projects/cabots/. The
views, opinions, and findings contained in this report are those of the author(s) and should not be construed as an
official National Oceanic and Atmospheric Administration or U.S. Government position, policy, or decision.

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

**Table 1. Summary of the lidar, surface, and aircraft comparisons**

| A | B | Ratio±1σ (A/B) | Diff.±1σ (A-B) | Slope* (A vs B) | Int.* (A vs B) |
|---|---|---|---|---|---|
| TOPAZ | VMA | 1.06±0.08 | 2.9±3.7 ppbv | 1.00±0.03 | -2.6±1.5 ppbv |
| SciAv | VMA | 1.07±0.10 | 5.0±5.0 ppbv | 1.01±0.01 | -4.5±1.1 ppbv |
| TOPAZ | SciAv | 1.01±0.04 | 0.8±2.8 ppbv | 1.00 ±0.13 | 1.0±9.0 ppbv |
| TOPAZ | AJAX | 1.08±0.06 | 4.2±0.8 ppbv | 1.07±0.13 | 1.8±3.4 ppbv |

*from Orthogonal Distance Regression (ODR) fits. Uncertainties are 95% confidence limits.

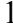

**Figure 1. (a) Topographic map showing the air basins of California (dashed black lines); the San Joaquin Valley Air**
**Basin (SJVAB) is outlined in heavy solid black. Interstate highways and urban areas are shown in gray. The filled red**
**triangles show the CABOTS measurement sites at Bodega Bay (BBY), Half Moon Bay (HMB), Visalia Municipal Airport**
**(VMA), and Chews Ridge Observatory (CRO).**


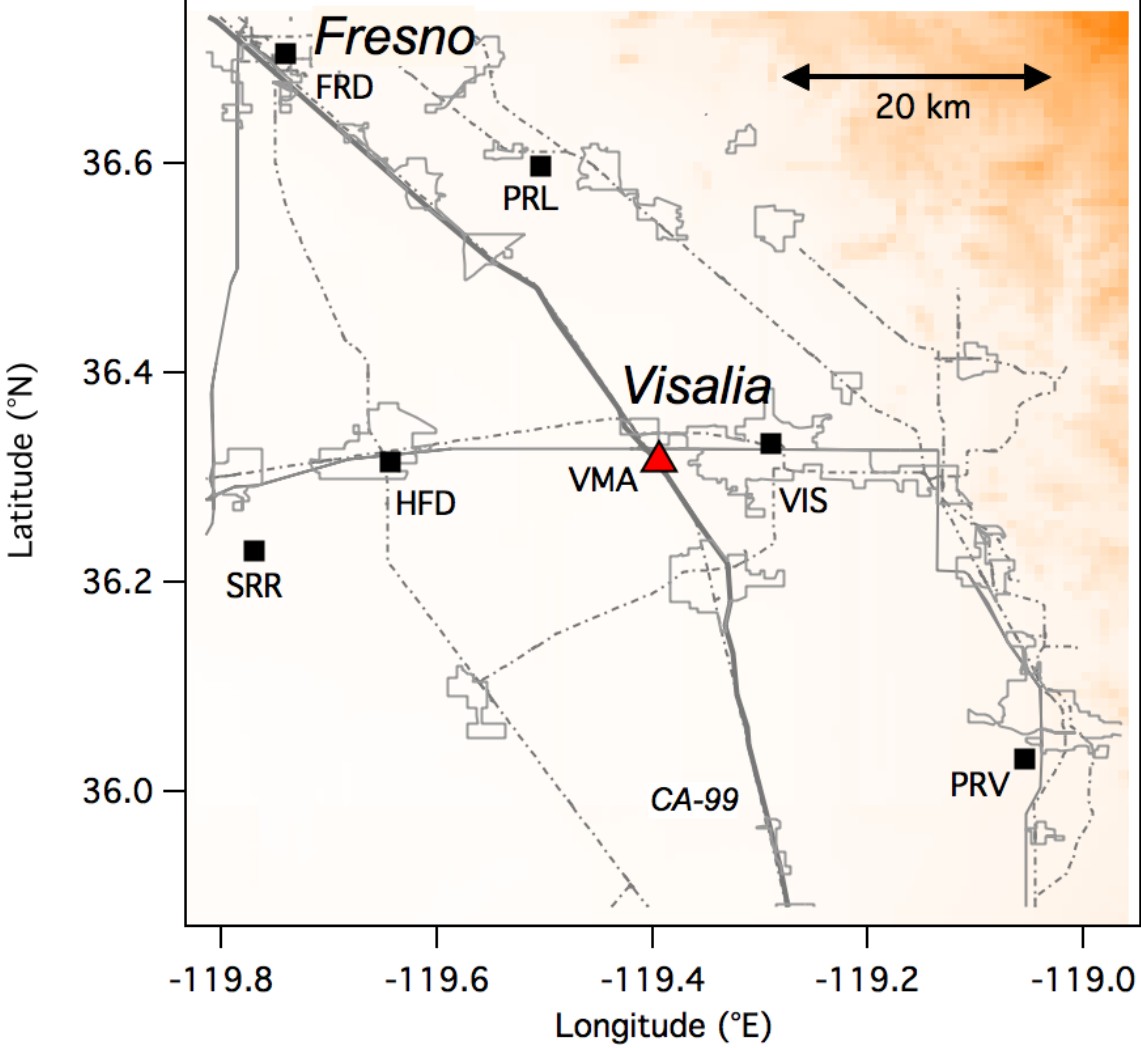

**Figure 1. (b) Same as (a), but showing an enlarged view of the area surrounding the VMA. The solid and dot-dash gray**
**lines represent the major highways and railroads, respectively, with the heavier solid line showing CA-99 (see text). The**
**filled black squares show the 6 closest regulatory O$_3$ monitors active during the CABOTS campaign: Visalia (VIS),**
**Hanford (HFD), Santa Rosa Rancheria (SRR), Fresno-Drummond St. (FRD), Parlier (PRL), and Porterville (PRV). The**
**elevation scale is the same as in (a).**

**Figure 2. Aerial view of the Visalia Municipal Airport (VMA) showing the 1 km lidar slant path line of sight as a yellow**
**arrow with the TOPAZ truck located at the base. The Scientific Aviation Mooney and AJAX Alpha Jet are shown**
**flanking the NOAA ESRL TOPAZ truck below the Google Earth image. Mooney and TOPAZ photos by A. Langford.**
**Alpha Jet photo by W. von Dauster.**

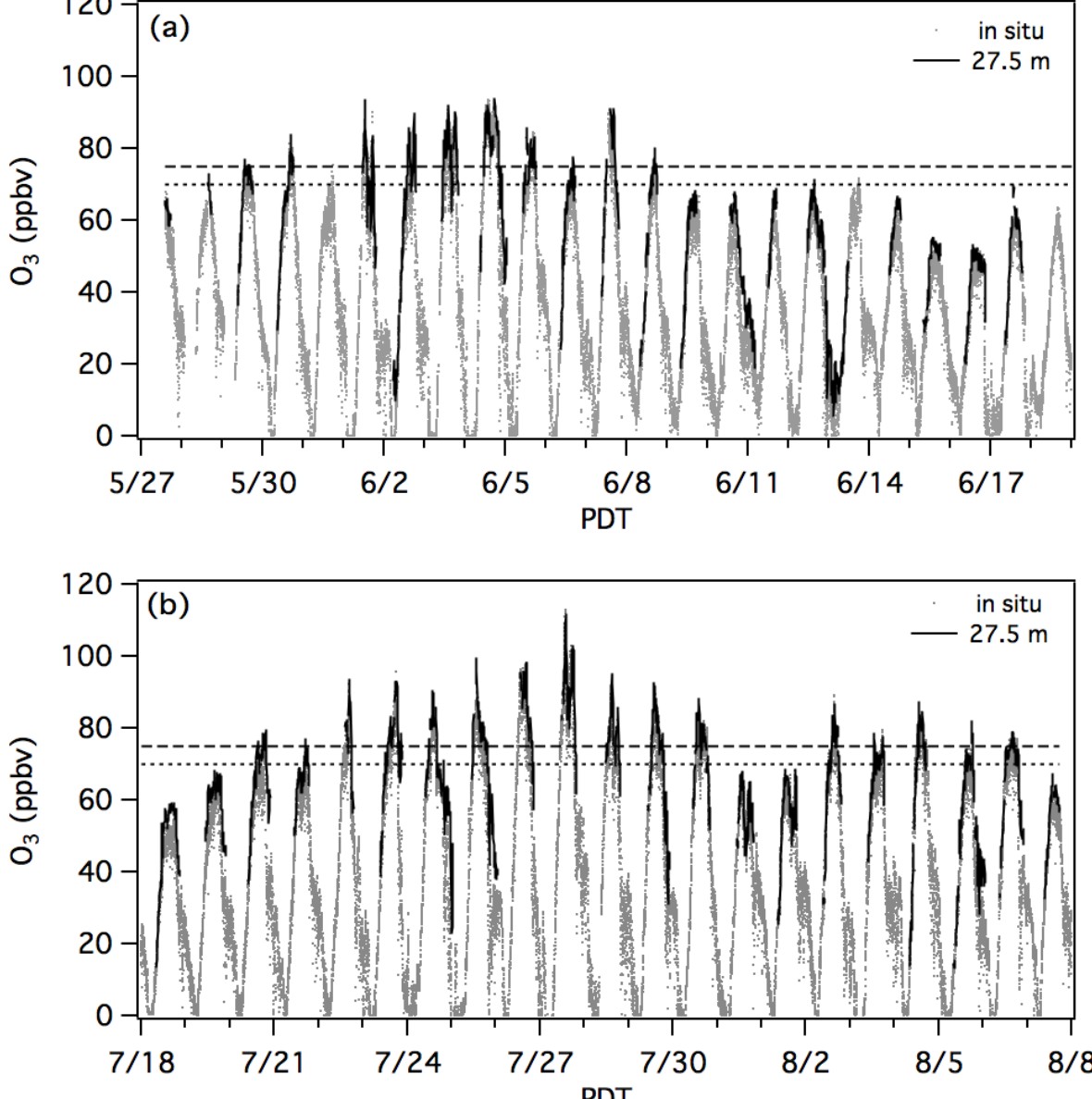

**Figure 3. Time series plots (local Pacific Daylight Time, PDT) of the O$_3$ concentrations retrieved 815±15 m downrange**
**and 27.5 m above the surface by TOPAZ (black line) with the measurements from the *in-situ* 2B monitor sampling 5 m**
**agl at the TOPAZ location (gray dots) during the first (a) and second (b) IOPs. The dashed and dotted lines respectively**
**show the 2008 (75 ppbv) and 2015 (70 ppbv) O$_3$ NAAQS.**

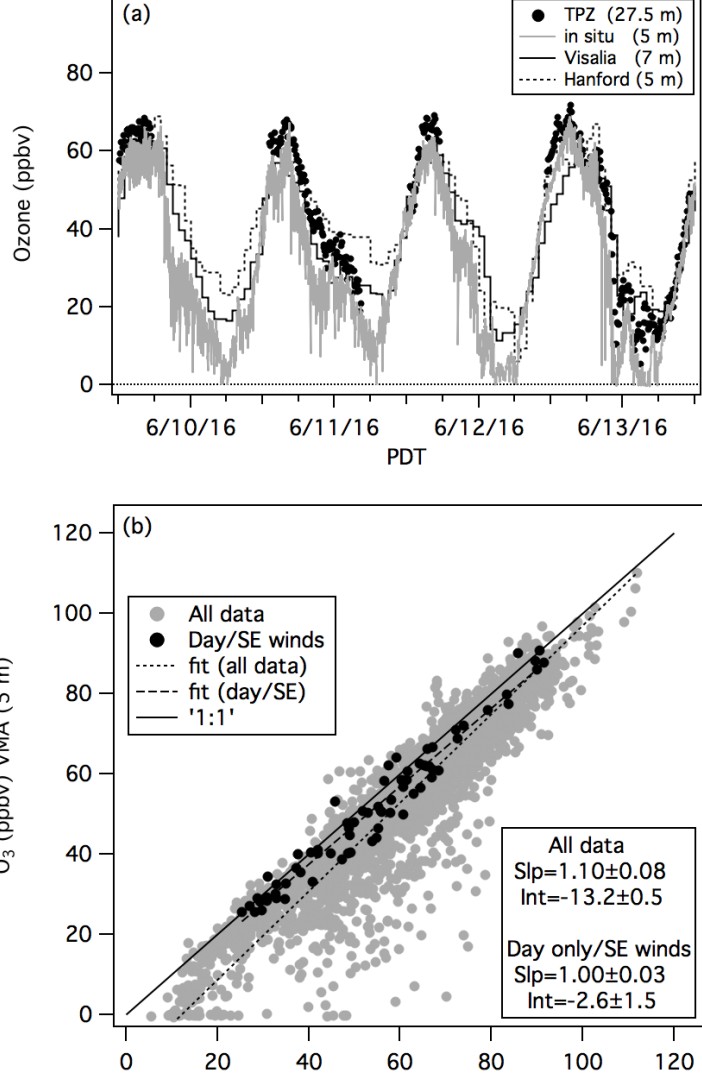

**Figure 4. (a) Four-day time series (9-13 June) showing the $O_3$ concentrations in air sampled 5 m agl above the TOPAZ truck at the VMA (gray line) and the $O_3$ mixing ratios at a height of 27.5±5 m and distance of 815±15 m retrieved from the TOPAZ measurements (filled black circles). The solid black and dotted staircase lines show the 1-h measurements from the Visalia and Hanford regulatory monitors. (b) Scatter plot comparing the 27.5 m TOPAZ measurements to the interpolated 5 m *in-situ* measurements. The filled gray circles (with dotted ODR fit) show the entire CABOTS data set from Figure 3, and the filled black circles (with dashed ODR fit) show only those measurements made during the day (0900 to 1830 PDT) when the winds were southeasterly (125 to 145°) and greater than 2.5 m s$^{-1}$. The solid line shows the 1:1 correspondence.**

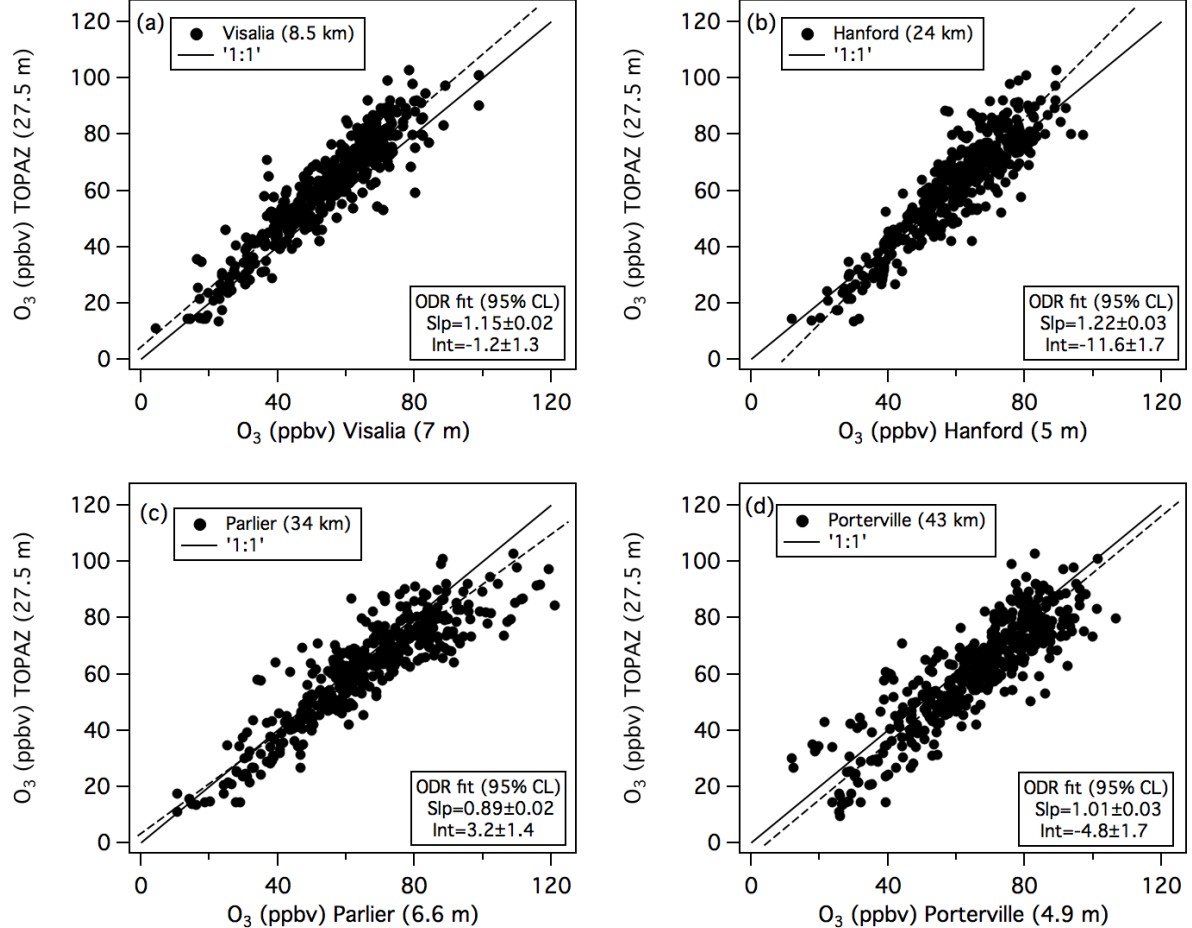

**Figure 5. Scatter plots with ODR fits comparing the 27.5 m TOPAZ measurements with the 1-h measurements from the regulatory monitors at (a) Visalia-N. Church Street, (b) Hanford, (c) Parlier, and (d) Porterville. The measurements in the upper box and x-axis label refer to the distance from the VMA and sampling height above ground, respectively. The Visalia monitor is operated by the California Resources Board. The remining three are operated by CARB and the SJVAPCD. The TOPAZ measurements are interpolated to the 1-h time base of the regulatory measurements for the comparison.**

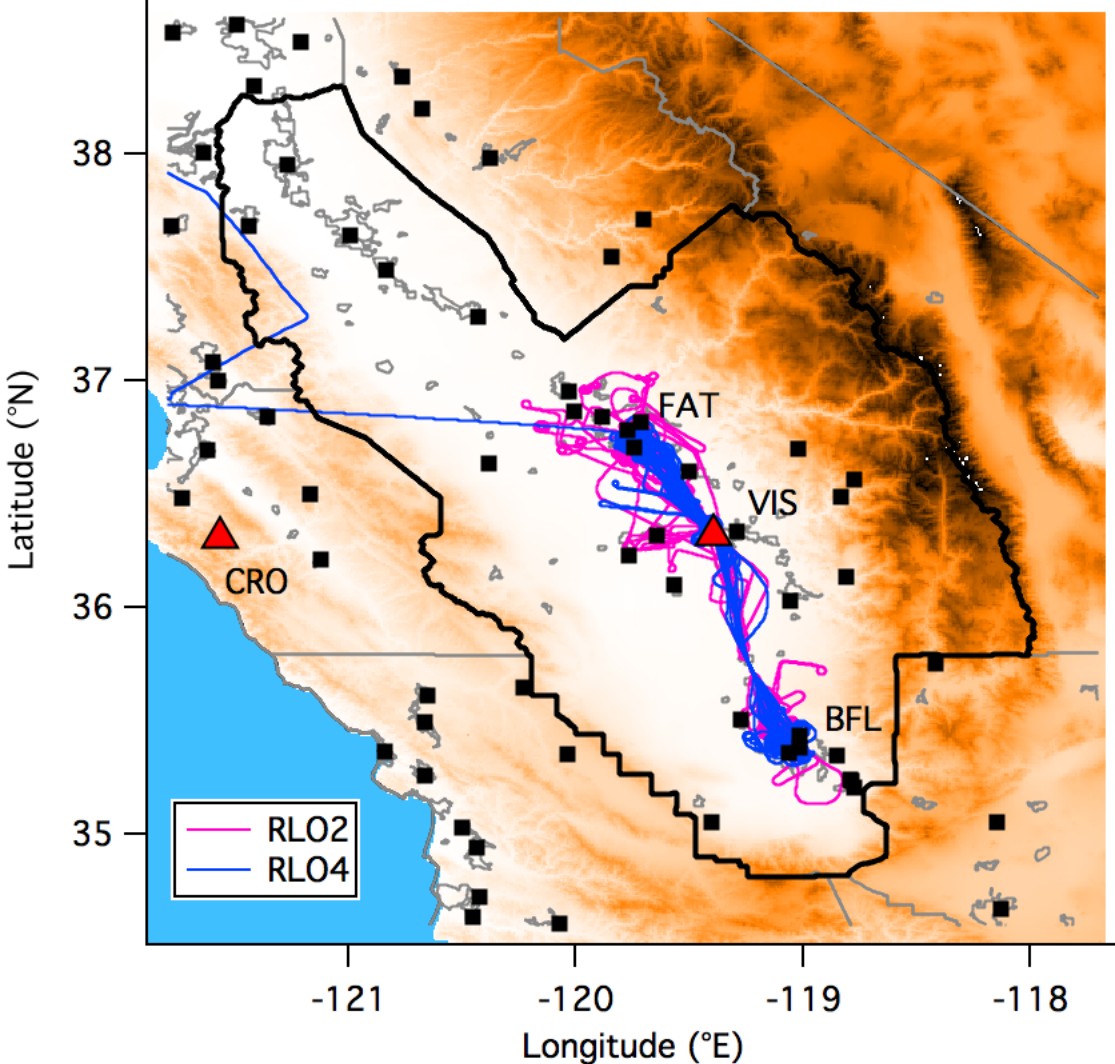

**Figure 6. (a) Map of the San Joaquin Valley showing the RLO flight tracks coincident with the TOPAZ measurements (RLO2 and RLO4). The filled black squares show the regulatory surface monitors. The CABOTS sampling sites at CRO and VMA are marked by red triangles. The other abbreviations are the Fresno (FAT), Visalia (VIS), and Bakersfield (BFL) airport codes. Note that VMA and VIS refer to the same airport.**


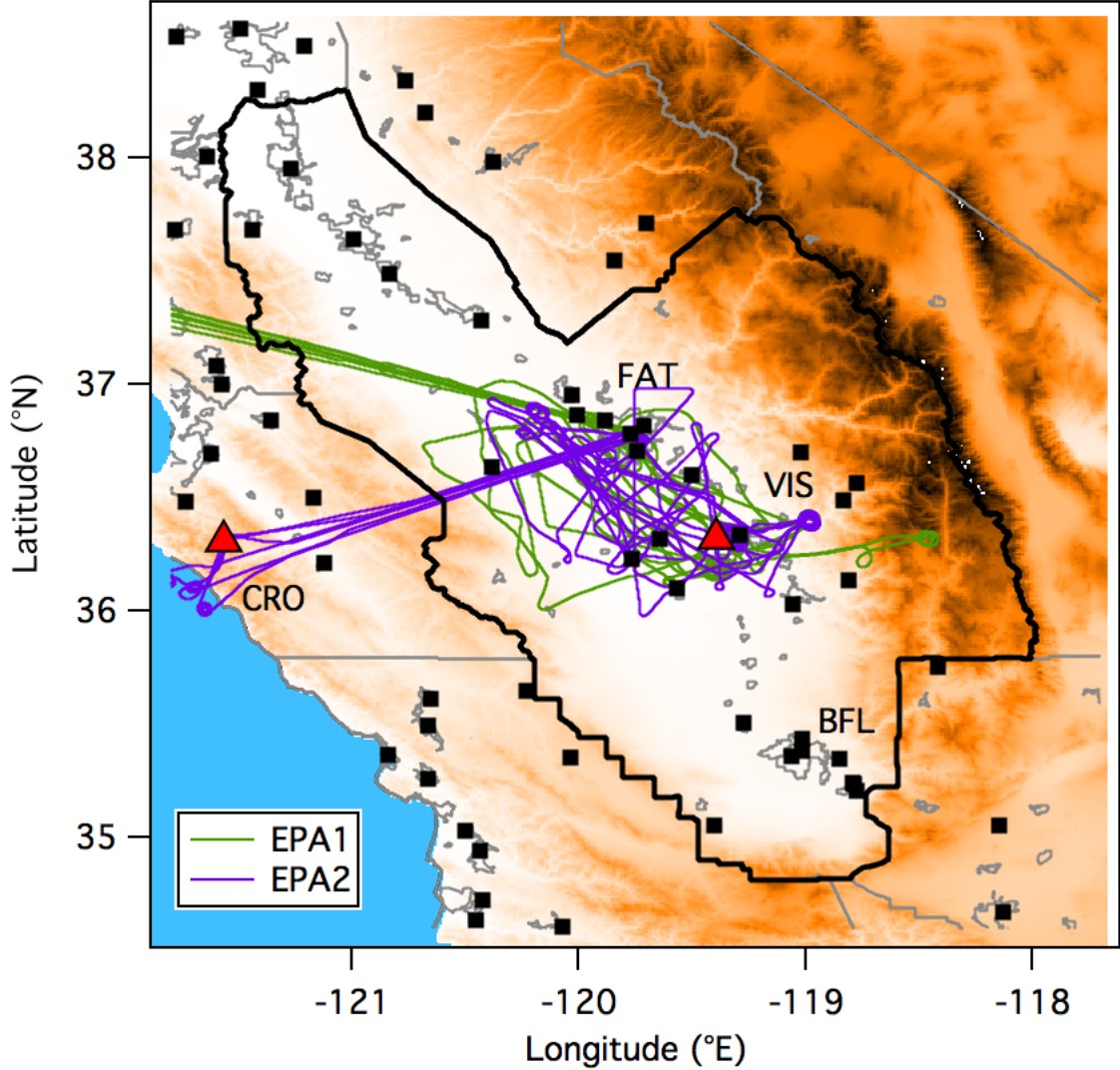

**Figure 6. (b) Same as (a), but with the EPA flight tracks (EPA1 and EPA2).**

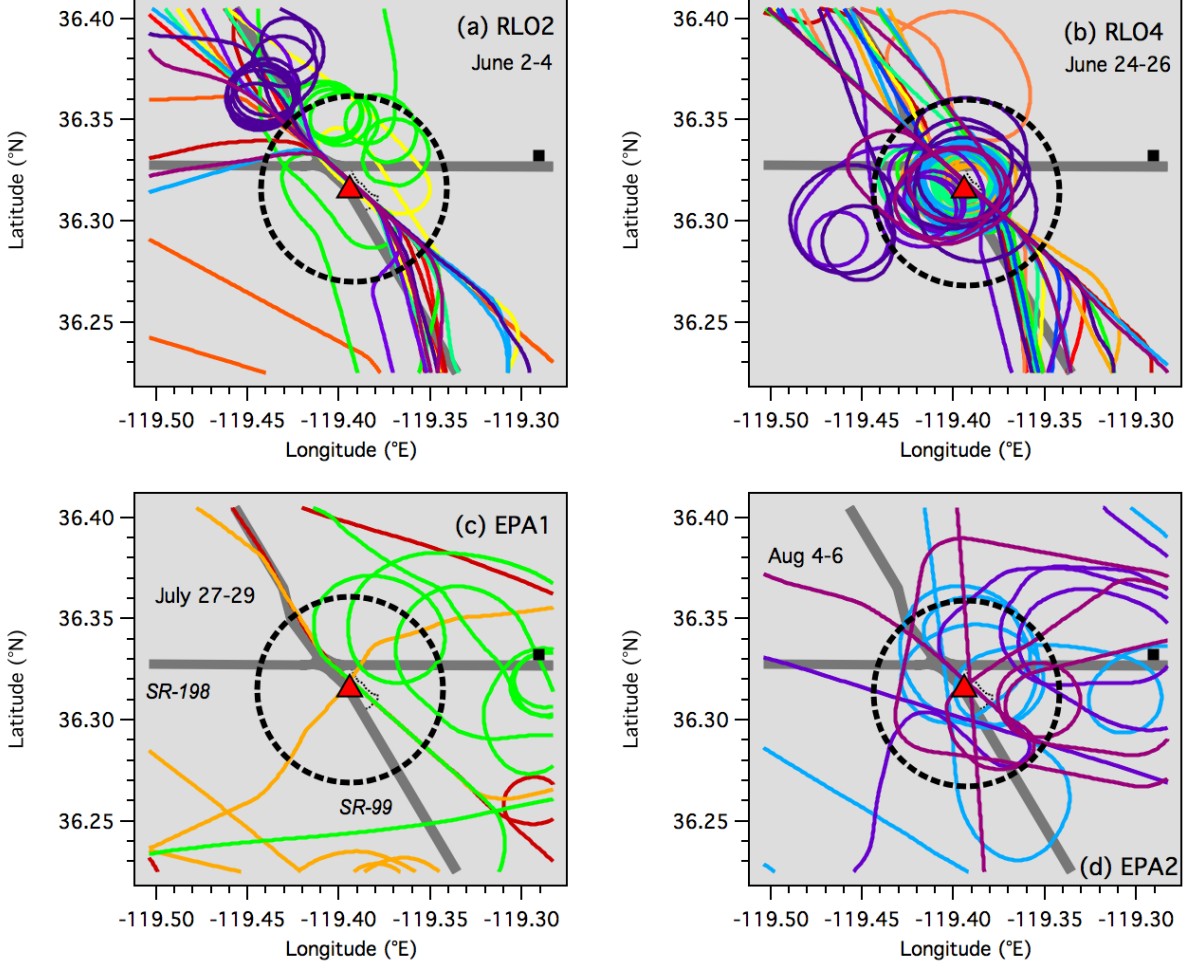

**Figure 7. RLO and EPA flight tracks in the vicinity of TOPAZ. (a) RLO2 (2-4 June), (b) RLO4 (24-26 July), (c) EPA1 (27-29 July), and (d) EPA2 (4-6 August). Each color represents a different flight. The red triangle marks the location of TOPAZ at the VMA and the dashed black circles show the 5 km radius used for the profile comparisons. The black square represents the Visalia-N. Church St. O₃ monitor.**

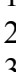

**Figure 8.** (a)-(d) Time series of the surface in-situ $O_3$ (gray dots) and 27.5 m TOPAZ $O_3$ (red line) measured during the RLO and EPA low approaches on (a) 2-5 June, (b) 24-27 July, (c) 27-30 July, and (d) 4-7 August 2016. The red envelope shows the the TOPAZ data ±3 ppbv, the nominal accuracy of the lidar retrievals. The blue squares represent the 1-s sampled (2-s recorded) Scientific Aviation measurements made between the surface and 25 m agl. The filled yellow circles in (a) and (c) show 2-s measurements from AJAX low approaches (see text). Panels (e) and (f) show scatter plots of the in-situ surface measurements and the Scientific Aviation data from the RLO flights in panels (a) and (b), respectively. The ODR fit parameters refer to the dark blue points which represent the measurements from daytime (0830-1830 PDT) flights.

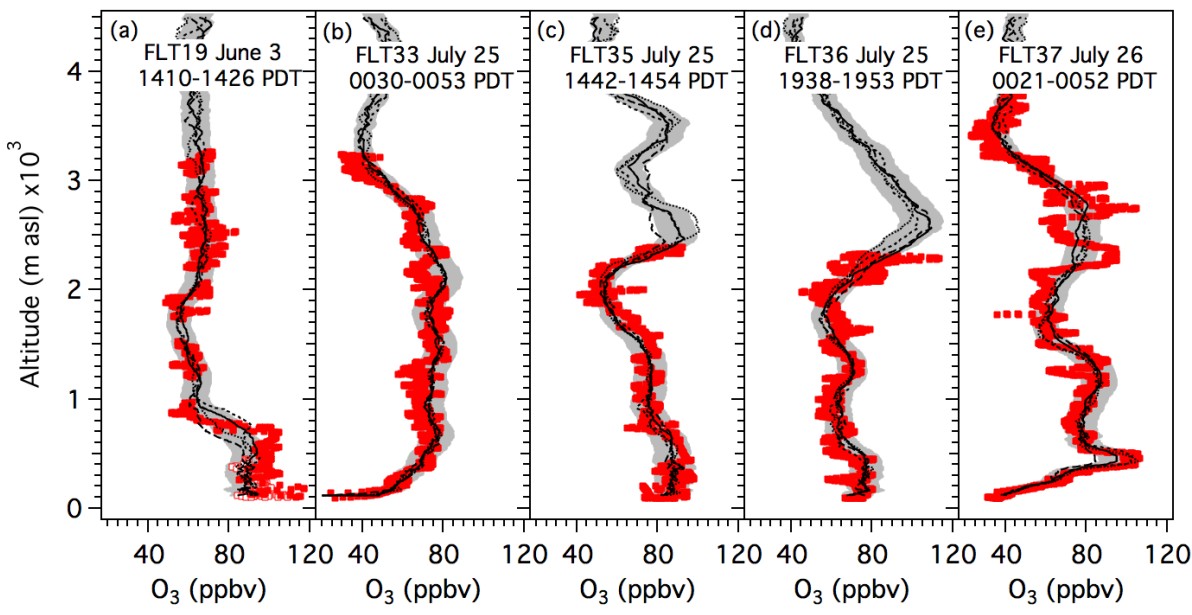

**Figure 9. Profile plots comparing the TOPAZ (black lines) and Scientific Aviation (red squares) O₃ measurements on (a) FLT19, 3 June, (b) FLT33, 25 July, (c) FLT35 25 July, (d) FLT36, 25 July and (e) FLT 37, 26 July. The dotted, short dash, solid, and long dash lines show the four consecutive 8-min lidar profiles acquired during the aircraft profiles. The gray envelopes show the mean lidar profile ±10% as reference. Note the large variability near the surface and sharp transition at 800 m in the 3 June aircraft measurements (cf. Figure 3a).**

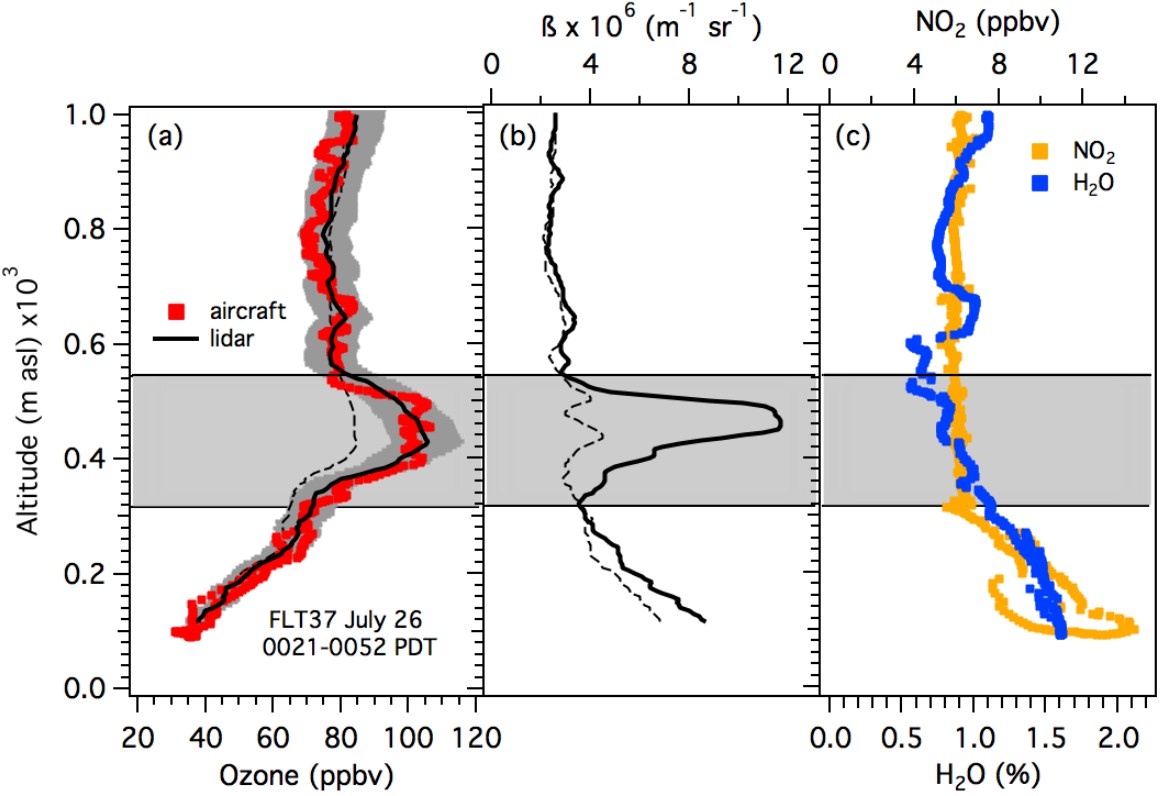

**Figure 10. (a) Expanded view of the lidar and aircraft O₃ profiles from Figure 9e plotted with coincident: (b) lidar backscatter, and (c) aircraft NO₂ and H₂O profiles. The solid black profile (±10% in gray) in (a) shows the lidar profile coinciding with the aircraft measurements below 1 km; the dashed black line shows the profile measured 16-24 minutes later. Likewise, for the backscatter profiles in (b). The horizontal gray band highlights the smoke puff from the Soberanes fire.**

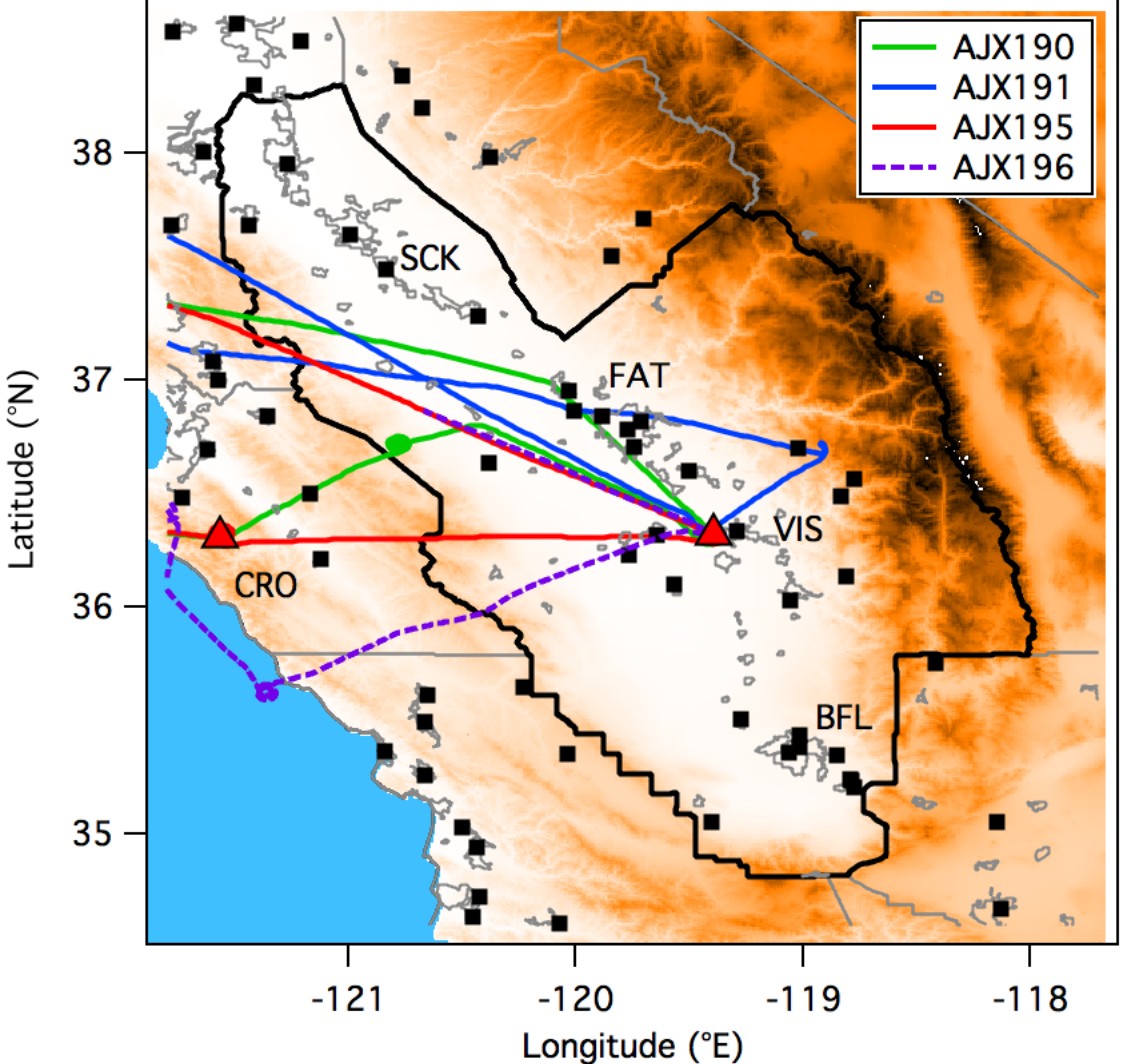

**Figure 11. Map of the San Joaquin Valley showing the AJAX flight tracks on 3 June (AJX190), 15 June (AJX191), 21 July (AJX195), and 28 July (AJX196). The abbreviations and symbols are the same as in Figure 6.**

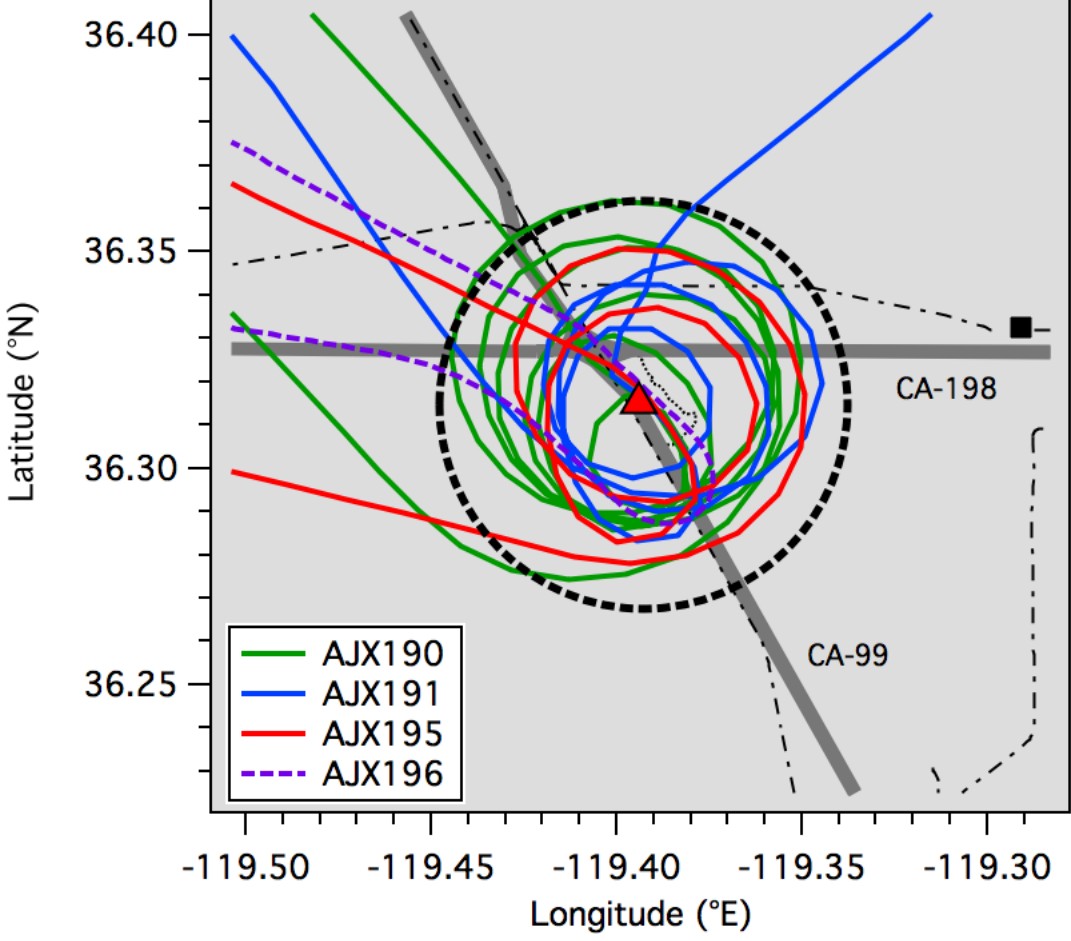

**Figure 12. AJAX flight tracks in the vicinity of the VMA (red triangle). The black square represents the Visalia-N. Church St. O$_3$ monitor and the dashed black circle marks the 5 km radius window used for the profile comparisons. The heavy gray lines show the major highways and the black dot-dash lines the railroads.**

2

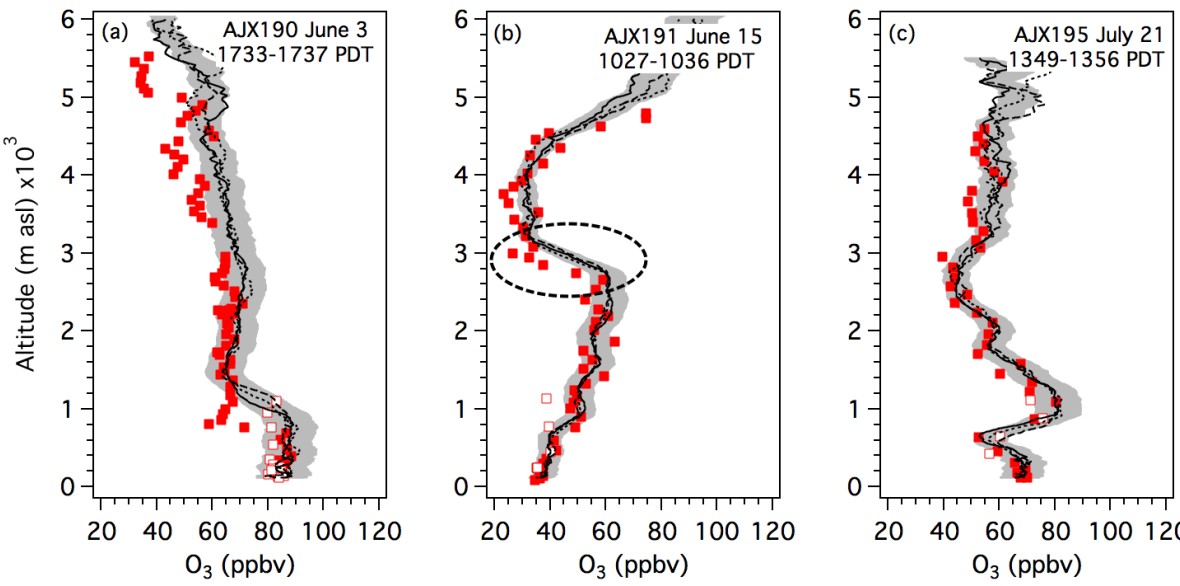

**Figure 13. Profile plots comparing the TOPAZ (black lines) and 10-s AJAX (red squares) measurements on (a) AJX190, 3**
**June, (b) AJX191, 15 June, and (c) AJX195, 21 July. The closed squares correspond to the Alpha Jet descent and the open**
**squares the subsequent climb out. Note the differences between these measurements. The dotted, dashed, and solid lines**
**show the order of the three 8-min lidar profiles that bracket the AJAX profile. The gray envelopes show the mean lidar**
**profile ±10% as reference. The significance of the dashed oval in (b) is discussed in the text.**

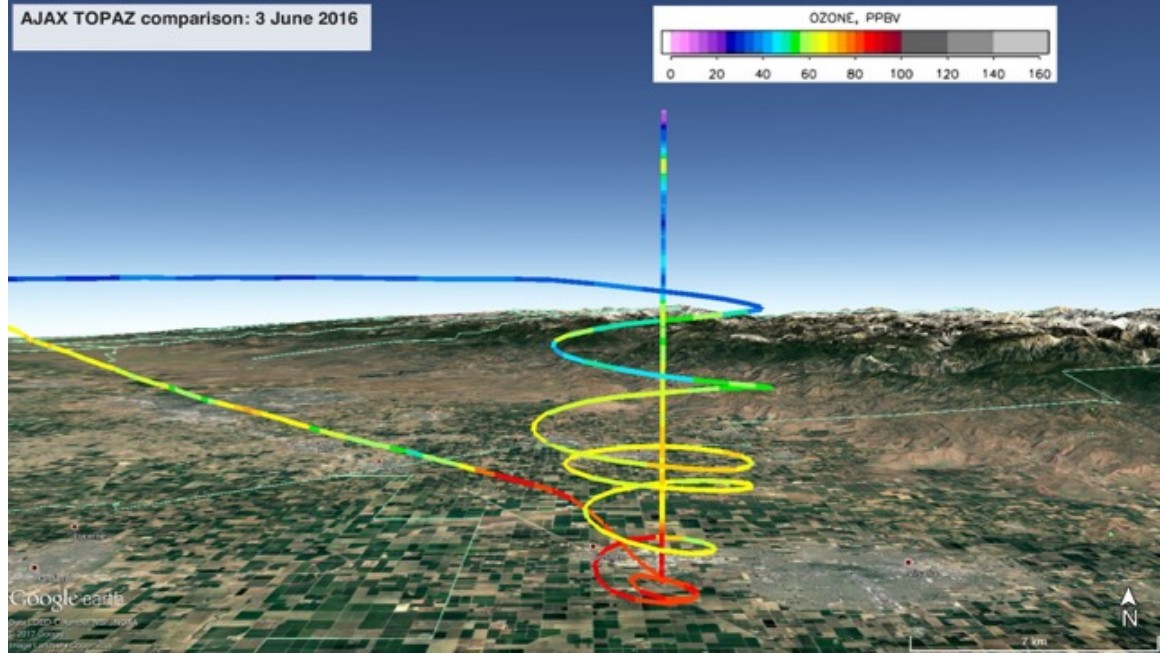

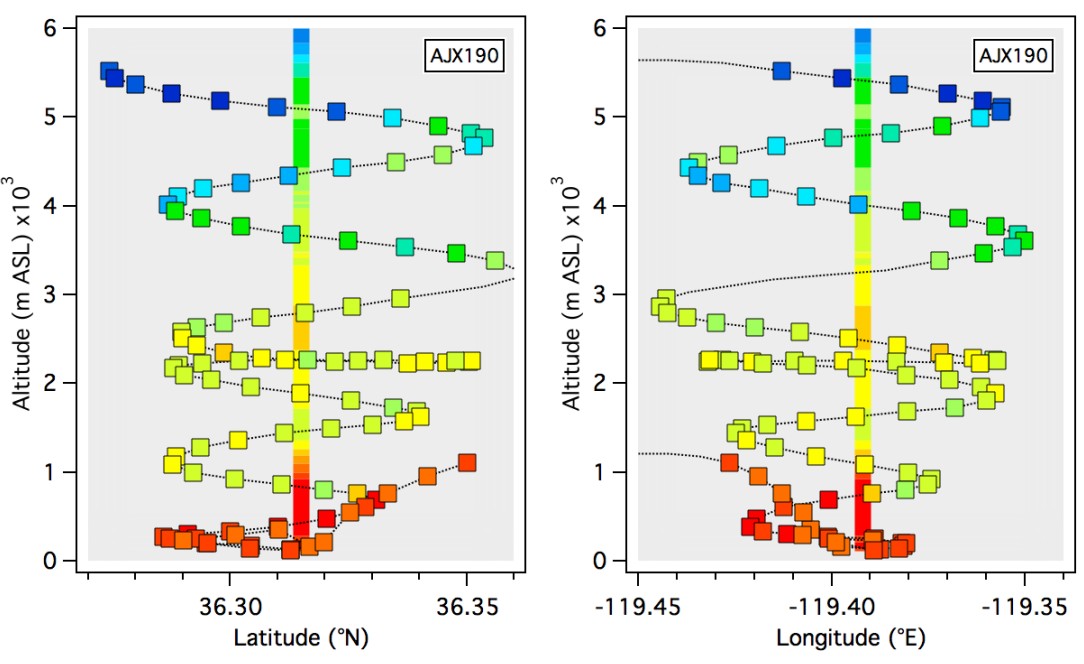

**Figure 14. (top) Google Earth image of the TOPAZ and AJAX profiles from 3 June 2016 showing the spatial variations**
**across the ~8 km diameter spiral profile by the Alpha Jet during its descent and climb out over the VMA. (bottom) AJAX**
**and TOPAZ profiles from Figure 13a plotted as a function of latitude (left) and longitude (right). Both plots are 10 km**
**wide. Note the strong horizontal gradients below 1.2 km.**

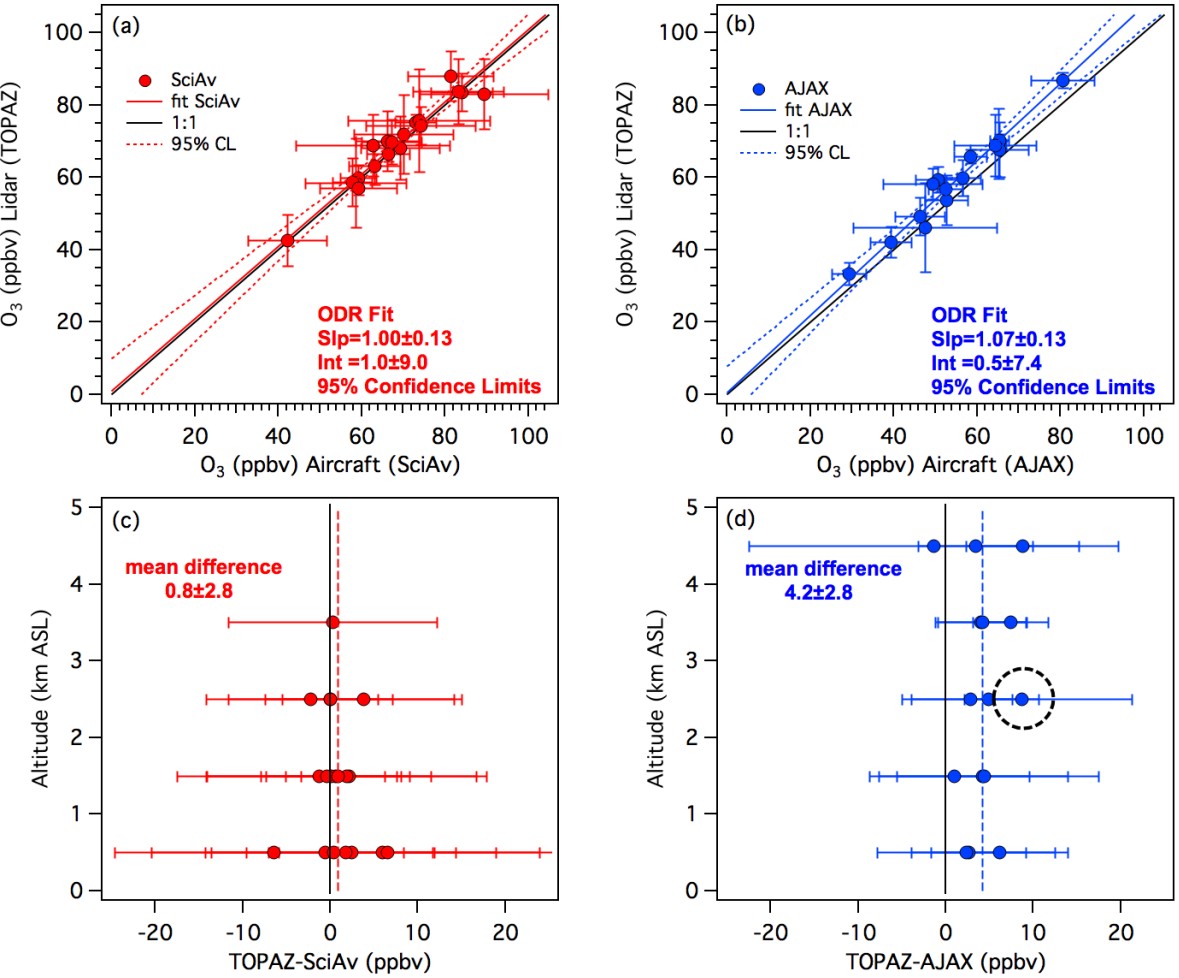

**Figure 15. (a) and (b), Scatter plots comparing the TOPAZ lidar retrievals to _in-situ_ O₃ measurements from 7 SciAv Mooney and 3 NASA Alpha Jet flights, respectively, averaged over 1 km vertical bins. The error bars show the standard deviations of the 1 km column means. (c) and (d), Differences between the 1 km mean TOPAZ and aircraft measurements from (a) and (b) plotted as a function of altitude. The vertical dashed lines show the mean differences. The dashed circle in (d) corresponds to the dashed oval in Figure 13b (see text).**

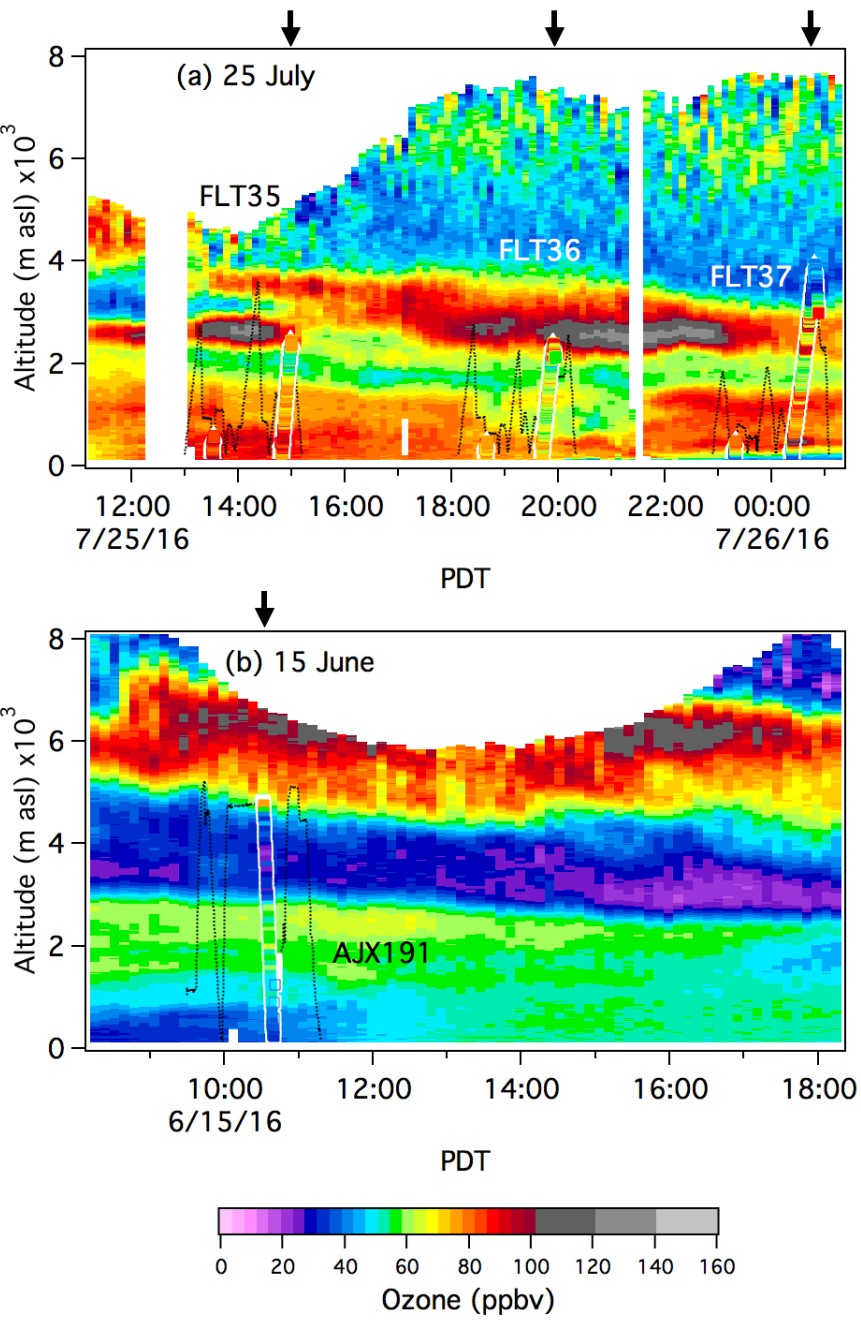

**Figure 16. Time-height curtain plots of the TOPAZ ozone measurements from (a) 25-26 July with the Scientific Aviation profiles from FLT35, 36, and 37 superimposed, and (b) 15 June with the coincident AJAX profile superimposed. The aircraft measurements made within 5 km of VMA (arrows) are highlighted by squares and colorized using the same scale as the TOPAZ data. The high O₃ layers around 3 km asl in (a) are related to the Soberanes Fire; the measurements plotted in the lower right corner of (a) correspond to the data shown in Figure 10.**