# Peer review of "Intercomparison of lidar, aircraft, and surface ozone measurements in the San Joaquin Valley during the California"

_Atmospheric Measurement Techniques, 2018_

## Referee Comment (RC1) · Anonymous Referee #1 · 27 Nov 2018

**Review of manuscript "Intercomparison of lidar, aircraft, and surface ozone measurements in the San Joaquin Valley during the California Baseline Ozone Transport Study (CABOTS)" by Langford et al.**

**General comments**

he paper deals with the analysis of comparisons between a UV ozone DIAL (Differential Absorption Lidar) and in-situ surface and aircraft measurements in Southern California during the CABOTS campaign. The objective is to assess differences be-

tween the ozone measurement techniques deployed during CABOTS. Authors focus on the comparibility of measurements close to the surface, thanks to the scanning capabilities of the lidar system. A detailed analysis of the aircraft flight path around the lidar is also essential to separate the role of horizontal variability from that of instrumental differences. Many studies have been published to compare UV DIAL with either aircraft, ozonesonde or surface observation. This is not enough recognized in the introduction where few reference to similar campaign are given. The topic is still sound and of interest for its publication in Atmospheric Measurement Techniques, because the comparison of lidar data with surface measurement is quite difficult. Specific aerosol interferences encountered in Southern California (biomass burning, polluted dust and advection of marine aerosols) also need to be characterized. However, the current paper need some minor revisions before its final publication in Atmospheric Measurement Techniques. My major concern is the lack of synthesis and quantitative discussion about the comparison results. The expected UV DIAL accuracy below 4 km presented in section 3.1 is $\pm$3 ppb, but this number is not compared with the differences observed during the campaign when taking into account the detailed analysis of the spatial variability conducted in the paper. Also the description of the aerosol interference correction is not very well explained although the ozone profiles shown in Fig. 10 prove that it is probably very efficient.

**Specific comments**

p.2 l.19 Ozone accuracy needed to address the stratospheric intrusion and long range transport studies could be given as a specification for the ozone measurement accuracy. Do you plan Lagrangian studies between aircraft and lidar observationsÂă?

p.2 l.24 Provide more references to previous lidar characterization using airborne or surface measurements (e.g. DOI: 10.1063/1.1144769, DOI: 10.1023/A:1021354511127, DOI: 10.1016/j.atmosres.2004.10.003, ...)

p.2 l.30 State in introduction the need for a good characterization of lidar retrieval be-

tween surface and 100 m, especially for pollution studies

p.4 l.15 Does the iterative technique include the aerosol interference correctionÂă? If yes specify the parameters used for this correction. Does it correspond to the aerosol encountered during CABOTSÂă?

p.4 l. 28 Specify the accuracy and detection limit for the Scientific Aviation (Sc.Av.) ozone monitor.

p. 5 l.6-8 Did you perform flights in formation between the two aircraft to identify differences between the two airborne in-situ ozone measurementsÂă? If yes please provide the range of observed O3 differences.

p.5 l. 37 *"the 27.5 m TOPAZ measurements were usually larger than the VMA in-situ"*. Specify the value of the bias. It seems larger than 3 ppb. Why not using all the measurement days shown in Fig. 3 to make the scatterplot in Fig. 4bÂă? The bias will be more representative, especially if the daytime and SE wind assumptions are included.

p.7 l.6 The sentence *"differences are within ±10%"* is not really useful if it is not detailed. Differences are hard to read in Fig. 8. Please specify bias for daytime. It is also useful to report on differences between aircraft and in-situ observations.

p.7 l.13 Why ±10% for the gray envelop and not the expected ±3 ppb lidar accuracyÂă?

p.7 l.16 *"The agreement between the TOPAZ and Mooney measurements in Figure 9 is quite good, with some notable differences"*. Specify how large are the differences or add scatterplot in addition to the vertical profile plots shown in Fig. 9.

p.7 l.22-26. The 10 ppb lidar underestimate in the 0-800 m altitude range on June 3rd is not really discussed while it is larger than differences observed for other flights in the same altitude range. What is the aerosol backscatter on this day ?

p.7 l.37 It is indeed an interesting comparison. Please give the parameters used for

the aerosol correction interference. Is it consistent with biomass burning aerosol optical properties in the UV ?

p.8 l.1 Please specify why you expect an interference from NO2 or H2O vertical distribution.

p.8 l.34 It is very difficult to see the magnitude of spatial ozone inhomogenieties in Fig. 14. Please give numbers or a better figure (x-z cross section along the dimension with largest horizontal gradient would be more explicit and easier to read).

p.9 l.2 The authors could show the 0.5-1.5 km range scatterplot in addition to the 1.5-2.5 km figure. Standart deviations may be larger for profiles with large ozone gradient, but the bias must remains small if the instrument accuracy is not the limiting factor. The issue of this paper is indeed to demonstrate that a good comparison is possible at range below 2 km.

p.9 l.11-13 Please make a quantitative summary of the comparison findings and discuss these numbers with the expected overall bias and single profile accuracy of the TOPAZ lidar.

---

## Referee Comment (RC2) · Anonymous Referee #2 · 27 Dec 2018

This paper describes the measurement campaign configuration of an extensive field campaign involving lidar, airborne in-situ and ground based in situ observations. The main purpose of the paper is to assess the data quality from the instruments during the campaign so that the data can be used for further process studies which are not described in the paper.

As is usual for large field campaigns, the set up is complex and involves many instruments (with different properties), operated at different sites or platforms (with consequently differing times and locations of observation). Taking this into account, the paper

is well organised and gives a clear view of the overall experiment. Some interpretation of the atmospheric chemistry cannot be avoided in order to interpret some of the differences observed, where perhaps better similarities would have been expected.

A few minor suggestions follow meant to improve the text. - 1. introduction - The first sentence mentions a 2016 design value, which is not easily understood. This sentence and concepts should be clarified. - 2. campaign design. Some of the abbreviations are rather long and awkward (i.e. SJVUAPCD) whereas in the figures all sites and instrument data are shortened to three letters. I suggest to shorten the unnecessarily long abbreviations and while at it harmonise with the labels and annotations in the figures. - 3.1 TOPAZ. Is it relevant to mention the changes to the instrument? Were this made since the last campaign and is this paper the source where these changes are documented? If not (i.e. reporting of changes has been done elsewhere) these details can be removed. - 3.1.pp4 line 9. A single sentence could be added to explain the expected effects of Nix emissions on measured ozone concentrations. - 3.2pp4 line 29. Explain why a Nox monitor with photolytic converter measuring NO and NO2 was sufficient and no NO2 specific instrument was used. - 4.1 comparison lidar surface. TOPAZ was compared to in-situ observations using a low elevation angle of the lidar and a distance of about 800 m along the profile. This results in a height above ground of about 27 m. The agreement with the corrected in-situ observations is good. However, the interval along the lidar profile at 800 m distance is only a small part of the full profile. Have there been attempts to validate/intercompare different ranges of the lidar profile with the ground based in-situ monitors? - 4.1 pp5 line 25 - I consider it a weak point that the TOPAZ truck was only equipped with an in-situ ozone monitor and no NOx of NO2 monitor. This would have been helpful since NO2 titration effects were expected in a polluted environment. Why was there no NOx/NO2 monitor? - 4.2.2 pp8 line 31. This sentence should probably be rearranged or split in two to clarify what was in agreement with what. - 5 summary pp9 line 25. Remove 'Although', add a full stop after 'with the lidar' and add 'However' before TOPAZ. This is to explain why the ozone sonde data has not been used in the intercomparison.

Figures - Fig. 3. mention the retrieval is lidar retrieval. Add the distance between the lidar volume and the location of the in-situ monitor. - Fig. 8. add in the caption the relevance of subfigures a,b,c and d.
* * *

---

## Author Comment (AC1) · 14 Feb 2019

**Responses to Comments by AMT_2018-338-RC1**

Note: We have included the reviewer comments in *italics* and our replies in **boldface** to aid the reader**.**

*The paper deals with the analysis of comparisons between a UV ozone DIAL (Differential Absorption Lidar) and in-situ surface and aircraft measurements in Southern California during the CABOTS campaign. The objective is to assess differences between the ozone measurement techniques deployed during CABOTS. Authors focus on the comparability of measurements close to the surface, thanks to the scanning capabilities of the lidar system. A detailed analysis of the aircraft flight path around the lidar is also essential to separate the role of horizontal variability from that of instrumental differences. Many studies have been published to compare UV DIAL with either aircraft, ozonesonde or surface observation.*

*This is not enough recognized in the introduction where few references to similar campaigns are given.*

*The topic is still sound and of interest for its publication in Atmospheric Measurement Techniques, because the comparison of lidar data with surface measurement is quite difficult.*

*Specific aerosol interferences encountered in Southern California (biomass burning, polluted dust and advection of marine aerosols) also need to be characterized. However, the current paper needs some minor revisions before its final publication in Atmospheric Measurement Techniques.*

*My major concern is the lack of synthesis and quantitative discussion about the comparison results. The expected UV DIAL accuracy below 4 km presented in section 3.1 is ±3 ppb, but this number is not compared with the differences observed during the campaign when taking into account the detailed analysis of the spatial variability conducted in the paper. Also, the description of the aerosol interference correction is not very well explained although the ozone profiles shown in Fig. 10 prove that it is probably very efficient.*

*Specific comments*

*p.2 l.19 Ozone accuracy needed to address the stratospheric intrusion and long range transport studies could be given as a specification for the ozone measurement accuracy.* **We have added a statement here referencing a desired accuracy as 10%, which is the nominal tropospheric accuracy of ECC ozonesondes.** *Do you plan Lagrangian studies between aircraft and lidar observations?* **Analysis of the CABOTS data including FLEXPART analyses is ongoing, but no Lagrangian studies directly linking the aircraft and lidar measurements are currently planned.**

*p.2 l.24 Provide more references to previous lidar characterization using airborne or surface measurements (e.g. DOI: 10.1063/1.1144769, DOI: 10.1023/A:1021354511127, DOI:*

*10.1016/j.atmosres.2004.10.003, ...).* **The suggested references have been added to the introduction.**

*p.2 l.30 State in introduction the need for a good characterization of lidar retrieval between surface and 100 m, especially for pollution studies.* **A statement to this effect has been added.**

*p.4 l.15 Does the iterative technique include the aerosol interference correction? If yes specify the parameters used for this correction. Does it correspond to the aerosol encountered during CABOTS?* **Yes, the iterative technique includes a correction for differential aerosol backscatter and extinction (this is described in detail in the Alvarez et al., 2011 reference) and a description of this process has been added to Section 3.1. We used Angstrom coefficients of 0 (no wavelength dependence) for aerosol backscatter and -0.5 for aerosol extinction for the entire study. These values are based on the work of Voelger et al. (1996) and are a good compromise for a wide variety of aerosol mixtures. The composition of the aerosols encountered during CABOTS was not directly measured, but wood smoke predominated in the latter part of the CABOTS campaign.**

*p.4 l. 28 Specify the accuracy and detection limit for the Scientific Aviation (ScAvi) ozone monitor.* **This information has been added to Section 3.2.**

*p. 5 l.6-8 Did you perform flights in formation between the two aircraft to identify differences between the two airborne in-situ ozone measurements? If yes please provide the range of observed O3 differences.* **There were no formation flights with the Mooney and Alpha Jet during CABOTS.**

*p.5 l. 37 "the 27.5 m TOPAZ measurements were usually larger than the VMA in- situ". Specify the value of the bias. It seems larger than 3 ppb. Why not using all the measurement days shown in Fig. 3 to make the scatterplot in Fig. 4b? The bias will be more representative, especially if the daytime and SE wind assumptions are included.* **The scatter plot in Fig. 4b *does* include all the measurement days and the bias (caused by localize titration) is in the in-situ measurements and *not* the lidar measurements. We have revised the text for greater clarity and the scatter plot axes in Fig. 4b have been reversed to emphasize this last point.**

*p.7 l.6 The sentence "differences are within ±10%" is not really useful if it is not detailed.* **See next comment.** *Differences are hard to read in Fig. 8. Please specify bias for daytime.* **There is no daytime bias in the TOPAZ measurements over the altitude ranges considered in this study.** *It is also useful to report on differences between aircraft and in-situ observations.* **Two scatter plots comparing the aircraft and in-situ measurements have been added to Figure 9. These data are consistent with the lidar-surface comparisons in Section 4.1.**

*p.7l.13 Why ±10% for the gray envelope and not the expected ±3 ppb lidar accuracy?* **Differences between the lidar and aircraft measurements include unknown contributions arising from the imperfect overlap of the sampled airmasses and the spatial and temporal variability of ozone. We use 10% as a reference because it is the nominal accuracy of ECC**

**ozonesondes, the generally accepted reference standard for ozone profiling, in the troposphere. We clarify this in text added to the beginning of Section 4.2.**

p.7 l.16 *"The agreement between the TOPAZ and Mooney measurements in Figure 9 is quite good, with some notable differences"* . Specify how large are the differences or add scatterplot in addition to the vertical profile plots shown in Fig. 9. The text has been expanded **Scatter plots showing altitude binned comparisons have been added to Figure 15 and a Table has been added to quantitatively summarize the differences between the measurements.**

*p.7 l.22-26. The 10 ppb lidar underestimate in the 0-800 m altitude range on June 3rd is not really discussed while it is larger than differences observed for other flights in the same altitude range. What is the aerosol backscatter on this day?* **The discussion of these measurements has been expanded and the red (SciAv) points in Figure 9 have been enlarged to better show that the lidar and aircraft measurements on 3 June do, in fact, overlap despite the large horizontal variability seen in the aircraft measurements. The aerosol loading on that day was not unusual.**

*p.7 l.37 It is indeed an interesting comparison. Please give the parameters used for the aerosol correction interference. Is it consistent with biomass burning aerosol optical properties in the UV ?* **See the comments for p.4 l.15 above.**

*p.8 l.1 Please specify why you expect an interference from NO2 or H2O vertical distribution.* **We do not *expect* an interference, but both NO$_2$ and H$_2$O have weak UV absorbances and the question of possible interferences in lidar retrievals has been considered in previous studies (see ref in text).**

*p.8 l.34 It is very difficult to see the magnitude of spatial ozone inhomogeneities in Fig. 14. Please give numbers or a better figure (x-z cross section along the dimension with largest horizontal gradient would be more explicit and easier to read).* **Two panels showing altitude-latitude and altitude-longitude plots have been added to the figure to help clarify this issue.**

*p.9 l.2 The authors could show the 0.5-1.5 km range scatterplot in addition to the 1.5- 2.5 km figure. Standard deviations may be larger for profiles with large ozone gradient, but the bias must remain small if the instrument accuracy is not the limiting factor. The issue of this paper is indeed to demonstrate that a good comparison is possible at range below 2 km.* **Figure 15 has been revised to show comparisons with data binned into 1 km intervals:0-1, 1-2, 2-3, 3-4, and 4-5 km (AJAX only).**

*p.9 l.11-13 Please make a quantitative summary of the comparison findings and discuss these numbers with the expected overall bias and single profile accuracy of the TOPAZ lidar.* **We have added a new discussion section (5) and a table to quantitatively summarize the comparison findings.**

---

## Author Comment (AC2) · 14 Feb 2019

**Responses to Comments by AMT_2018-338-RC2**

Note: We have included the reviewer comments in *italics* and our replies in **boldface** to aid the reader**.**

*This paper describes the measurement campaign configuration of an extensive field campaign involving lidar, airborne in-situ and ground based in situ observations. The main purpose of the paper is to assess the data quality from the instruments during the campaign so that the data can be used for further process studies which are not described in the paper.*

*As is usual for large field campaigns, the setup is complex and involves many instruments (with different properties), operated at different sites or platforms (with consequently differing times and locations of observation). Taking this into account, the paper is well organised and gives a clear view of the overall experiment. Some interpretation of the atmospheric chemistry cannot be avoided in order to interpret some of the differences observed, where perhaps better similarities would have been expected.*

*A few minor suggestions follow meant to improve the text.*

*1. introduction - The first sentence mentions a 2016 design value, which is not easily understood. This sentence and concepts should be clarified.* – **This sentence has been revised to clarify the significance of the ozone Design Value.**

*2. campaign design. Some of the abbreviations are rather long and awkward (i.e. SJVUAPCD) whereas in the figures all sites and instrument data are shortened to three letters. I suggest to shorten the unnecessarily long abbreviations and while at it harmonise with the labels and annotations in the figures.* –**SJVUAPCD was shortened to SJVAPCD in the text and the "EPA/BAAQMD" flights are now simply labelled "EPA" in the text and figures.**

*3.1 TOPAZ. Is it relevant to mention the changes to the instrument? Were this made since the last campaign and is this paper the source where these changes are documented? If not (i.e. reporting of changes has been done elsewhere) these details can be removed.* – **This section has been revised, but we have retained many of the details since the ground-based version of TOPAZ has not been described in a dedicated instrument paper, and we wish to emphasis the differences between the instrumentation in this study and that used in other field campaigns described in the literature.**

*3.1.pp4 line 9. A single sentence could be added to explain the expected effects of NOx emissions on measured ozone concentrations.* – **The last sentence was expanded to incorporate this suggestion.**

*3.2pp4 line 29. Explain why a NOx monitor with photolytic converter measuring NO and NO2 was sufficient and no NO2 specific instrument was used.* – **The single engine Mooney aircraft has a limited payload and could not accommodate a more extensive instrumentation suite.**

*4.1 comparison lidar surface. TOPAZ was compared to in-situ observations using a low elevation angle of the lidar and a distance of about 800 m along the profile. This results in a height above ground of about 27 m. The agreement with the corrected in-situ observations is good. However, the interval along the lidar profile at 800 m distance is only a small part of the full profile. Have there been attempts to validate/intercompare different ranges of the lidar profile with the ground based in-situ monitors?* – **No.**

*4.1 pp5 line 25 - I consider it a weak point that the TOPAZ truck was only equipped with an in-situ ozone monitor and no NOx of NO2 monitor. This would have been helpful since NO2 titration effects were expected in a polluted environment. Why was there no NOx/NO2 monitor?* – **The CABOTS field experiment was designed to characterize the distribution of ozone aloft and not the photochemical state at the surface. The limited resources were allotted accordingly.**

*4.2.2 pp8 line 31. This sentence should probably be rearranged or split in two to clarify what was in agreement with what.* – **The sentence has been revised for clarity.**

*5 summary pp9 line 25. Remove 'Although', add a full stop after 'with the lidar' and add 'However' before TOPAZ. This is to explain why the ozone sonde data has not been used in the intercomparison.* **The suggested changes have been made in the text. We have also added statements to this effect in Section 2.**

*Figures - Fig. 3. mention the retrieval is lidar retrieval. Add the distance between the lidar volume and the location of the in-situ monitor. - Fig. 8. add in the caption the relevance of subfigures a,b,c and d.* **The suggested changes have been made and two additional panels have been added.**